# Theranostic Potentials of Gold Nanomaterials in Hematological Malignancies

**DOI:** 10.3390/cancers14133047

**Published:** 2022-06-21

**Authors:** Md Salman Shakil, Mahruba Sultana Niloy, Kazi Mustafa Mahmud, Mohammad Amjad Kamal, Md Asiful Islam

**Affiliations:** 1Department of Mathematics and Natural Sciences, Brac University, 66 Mohakhali, Dhaka 1212, Bangladesh; 2Department of Pharmacology & Toxicology, University of Otago, Dunedin 9016, New Zealand; 3Department of Biochemistry and Molecular Biology, Jahangirnagar University, Savar, Dhaka 1342, Bangladesh; mahruba.niloy@gmail.com (M.S.N.); kazi.stu2012@juniv.edu (K.M.M.); 4Institutes for Systems Genetics, Frontiers Science Center for Disease-Related Molecular Network, West China Hospital, Sichuan University, Chengdu 610064, China; prof.ma.kamal@gmail.com; 5King Fahd Medical Research Center, King Abdulaziz University, Jeddah 21589, Saudi Arabia; 6Department of Pharmacy, Faculty of Allied Health Sciences, Daffodil International University, Dhaka 1207, Bangladesh; 7Enzymoics, 7 Peterlee Place, Novel Global Community Educational Foundation, Hebersham, NSW 2770, Australia; 8Department of Haematology, School of Medical Sciences, Universiti Sains Malaysia, Kubang Kerian 16150, Kelantan, Malaysia; 9Institute of Metabolism and Systems Research, University of Birmingham, Birmingham B15 2TT, UK

**Keywords:** gold nanomaterials, hematological malignancies, diagnosis, treatment, promise and challenges

## Abstract

**Simple Summary:**

Hematological malignancies (HMs) cover 50% of all malignancies, and people of all ages can be affected by these deadly diseases. In many cases, conventional diagnostic tools fail to diagnose HMs at an early stage, due to heterogeneity and the long-term indolent phase of HMs. Therefore, many patients start their treatment at the late stage of HMs and have poor survival. Gold nanomaterials (GNMs) have shown promise as a cancer theranostic agent. GNMs are 1 nm to 100 nm materials having magnetic resonance and surface-plasmon-resonance properties. GNMs conjugated with antibodies, nucleic acids, peptides, photosensitizers, chemotherapeutic drugs, synthetic-drug candidates, bioactive compounds, and other theranostic biomolecules may enhance the efficacy and efficiency of both traditional and advanced theranostic approaches to combat HMs.

**Abstract:**

Hematological malignancies (HMs) are a heterogeneous group of blood neoplasia generally characterized by abnormal blood-cell production. Detection of HMs-specific molecular biomarkers (e.g., surface antigens, nucleic acid, and proteomic biomarkers) is crucial in determining clinical states and monitoring disease progression. Early diagnosis of HMs, followed by an effective treatment, can remarkably extend overall survival of patients. However, traditional and advanced HMs’ diagnostic strategies still lack selectivity and sensitivity. More importantly, commercially available chemotherapeutic drugs are losing their efficacy due to adverse effects, and many patients develop resistance against these drugs. To overcome these limitations, the development of novel potent and reliable theranostic agents is urgently needed to diagnose and combat HMs at an early stage. Recently, gold nanomaterials (GNMs) have shown promise in the diagnosis and treatment of HMs. Magnetic resonance and the surface-plasmon-resonance properties of GNMs have made them a suitable candidate in the diagnosis of HMs via magnetic-resonance imaging and colorimetric or electrochemical sensing of cancer-specific biomarkers. Furthermore, GNMs-based photodynamic therapy, photothermal therapy, radiation therapy, and targeted drug delivery enhanced the selectivity and efficacy of anticancer drugs or drug candidates. Therefore, surface-tuned GNMs could be used as sensitive, reliable, and accurate early HMs, metastatic HMs, and MRD-detection tools, as well as selective, potent anticancer agents. However, GNMs may induce endothelial leakage to exacerbate cancer metastasis. Studies using clinical patient samples, patient-derived HMs models, or healthy-animal models could give a precise idea about their theranostic potential as well as biocompatibility. The present review will investigate the theranostic potential of vectorized GNMs in HMs and future challenges before clinical theranostic applications in HMs.

## 1. Introduction

Hematological malignancies (HMs) are a heterogeneous group of blood neoplasms that are different from solid tumors, especially because of the presence of bone marrow (BM) suppression or failure symptoms [1]. In 2020, about 1.3 million people were diagnosed with HMs, while 0.7 million died due to HMs [2]. HMs comprise about 50% of all malignancies in children and 5%–8% of adult malignancies [3]. The pathology and physiology of HMs, irrespective of their indolent or aggressive stage, affect patients of all ages [4]. Therefore, early-diagnosis techniques and effective therapeutic regimes are needed to tackle this deadly disease.

Advanced molecular-biology techniques, including polymerase chain reaction (PCR), flow cytometry (FC), fluorescence in situ hybridization, immunocytochemical, immunophenotyping, karyotype analysis, and next-generation sequencing (NGS), offer a radical improvement in the diagnosis of HMs [4,5]. Among these techniques, NGS, PCR, and FC are the most commonly used to diagnose minimal-residual disease (MRD), a condition when residual HMs are present [6,7,8]. For example, PCR and FC techniques can be used to detect B cell immunoglobulin (lg) and its rearrangement to monitor disease progression as well as exosome-based MRD identification [9,10]. Similarly, significant advancement has been achieved in the treatment of HMs [11,12,13]. Immunotherapy, radioimmunotherapy, radiotherapy, chemotherapy, hematopoietic-cell transplantation, and BM transplantation improve the survival of patients with HMs [13,14,15,16]. However, due to the heterogeneity and long-term indolent phase, early diagnosis and treatment of HMs remain a challenging issue [12,17,18,19]. Additionally, following therapy, accurate diagnosis of MRD sometimes could not be possible due to false-negative or false-positive results from PCR, FC, and NGS tests [20,21,22]. Furthermore, false-negative results from FC were reported as false-positive in PCR in the diagnosis of MRD [23]. Therefore, sensitive and reliable HMs-diagnosis methods are still in demand, especially when a deficient level of MRD biomarker is present in the test sample [24]. Likewise, novel therapeutic approaches are urgently needed for HMs to overcome drug resistance, prevent cancer relapse, reduce side effects, improve selectivity and efficacy of currently used drugs, and, finally, enhance patient survival and quality of life [11,19,25,26].

Gold nanomaterials (GNMs) are 1–100 nm-sized nanomaterials of different shapes (e.g., nanorods, nanospheres, and nanocubes) composed of gold (Figure 1) [27,28]. Recently, GNMs have drawn considerable attention as cancer theranostic agents, a term that combines cancer diagnosis with therapy [27,29]. In diagnosis, the binding event between GNMs and analytes can change the physicochemical properties of GNMs, such as magnetic resonance (MR), surface-plasmon resonance (SPR), redox behavior, and conductivity, leading to distinguishable signals [30,31]. Due to the MR and SPR phenomenon, GNMs have become attractive candidates for magnetic-resonance imaging (MRI), colorimetric sensor, electrochemical sensor, and many cancer diagnoses [28,30,32]. Similarly, due to their unique properties and high surface area, GNMs function as practical platforms for binding multifunctional therapeutic moieties, including drugs, nucleic acids, peptides, and targeting biomolecules [30,33]. GNMs modified with therapeutic agents can be applied for a range of therapeutic applications including photodynamic therapy (PDT), photothermal therapy (PTT), radiation therapy (RDT), targeted drug delivery, and more [27,28]. Furthermore, GNMs can improve the efficacy and minimize the side effects of chemotherapeutic drugs by selective targeting [32]. Therefore, GNMs conjugation may improve the selectivity index of chemotherapeutic drugs, making them more biocompatible to normal healthy cells [27,32].

Recently, Huang et al. (2022) [34] reported on the applications of metal nanomaterials in the diagnosis and treatment of HMs. However, detailed discussions on GNMs were not observed in that review [34]. In the current review, we included research articles published before February 2021 that covered our topics of interest (i.e., GNMs and HMs). “Gold” and “leukemia”, “leukemias”, “leukaemia”, “leukaemias”, “lymphoma”, “lymphomas”, “myeloma”, “myelofibrosis”, “polycythemia vera”, “thrombocythemia”, “myelodysplastic syndromes”, and “myeloproliferative neoplasms” keywords were searched in the article title from four scientific databases (Google Scholar, Scopus, Web of Science, and PubMed).

**Figure 1 cancers-14-03047-f001:**
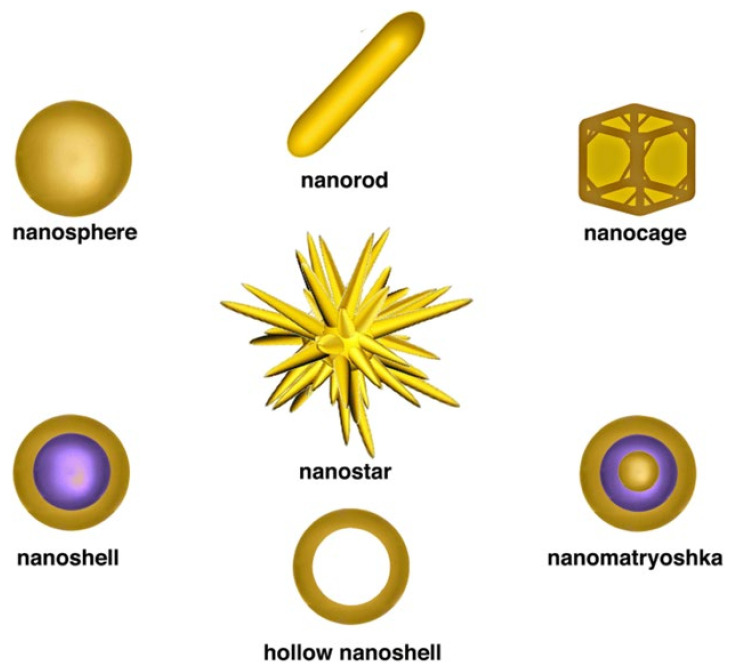
Different shapes of gold nanomaterials. The figure is reprinted from Berardis et al. (2021) [35]. This study is under Creative Commons Attribution 4.0 International License, which permits use, sharing, adaptation, distribution, and reproduction in any medium or format, as long as you give appropriate credit to the original author(s) and the source as well as provide a link to the Creative Commons license (http://creativecommons.org/licenses/by/4.0/, accessed on 26 May 2022).

In this review, we scrutinized the promise and challenges of using GNMs in the diagnosis and treatment of HMs. As a diagnostic agent, their accuracy, reliability, and sensitivity were investigated along with their therapeutic efficacy, potency, selectivity, mechanism of action(s), biocompatibility, and therapeutic limitations.

## 2. Hematological Malignancies

HMs are a group of heterogeneous diseases of diverse incidence, etiology, and prognosis [36]. Based on population-based studies, HMs are grouped into three broad categories: lymphoma, leukemia, and myeloma [2,37]. Among these three classes, lymphoma and leukemia are more prevalent HMs (Figure 2) [2]. Overall, the incidence of HMs appears to be increasing, but the epidemiology of HMs is unpredictable, mainly in Europe [37,38].

Genetic or epigenetic changes within normal hematopoietic cells turn them into malignant hematopoietic cells through dysregulation of proliferation, differentiation, and self-renewal (Figure 3) [41,42,43]. Three types of genes are linked with the etiopathogenesis of HMs: tumor-suppressor genes, oncogenes, and genes that offer genome stability [43]. Identification of the mutated genes or their molecular changes is helpful in differentiating histoclinical changes, therapeutic target sections, monitoring therapeutic effects, and disease progression [43,44]. Furthermore, the cBioPortal (https://www.cbioportal.org/, accessed on 29 January 2022) database [45,46] has shown that gene-mutation frequency varies in different HMs classes. Figure 1b,c illustrates the top ten mutations in leukemia and lymphoma patients according to the cBioPortal, which calculated the published data on leukemia [47,48,49,50,51,52,53,54,55,56] and lymphoma [57,58,59,60,61,62,63,64,65,66,67,68,69,70].

## 3. Gold-Nanomaterial-Based Diagnosis

GNMs-based molecular diagnosis of HMs can be placed into the following categories: (1) detection of leukemia or lymphoma cells based on unique antigen-receptor genes of T- and B-cells, (2) detection of genetic mutation(s), (3) identification of chromosomal deletions, translocations, or duplications, and (4) proteomic changes [71,72,73,74]. This section will report the accuracy, specificity, and reliability of aptamer, biocompatible polymer, antibody, nucleic acid, or ligand-tuned GNMs in the diagnosis of HMs.

### 3.1. Detection of Cancer Cells

Haghighi et al., (2020) [71] reported that gold nanocluster (GNC) modified with Fe3O4 and KH1C12 aptamer selectively diagnosed highly malignant HL-60 human-leukemia cells from a mixture of different cells. This nanoprobe could be used in MRI imaging (T_2_-based) or fluorescent-microscopy-based diagnosis of leukemia cells (as low as 10 cells μL^−1^). The fluorescent signal increased with an increase in nanoprobe concentration, while such an increase in the nanoprobe decreased the T_2_-based contrast. A major limitation of this study was no normal or cancerous cell line of blood origin was used as control. Only HepG2, liver-cancer cells were used as control (KH1C12-negative cells). Additionally, the authors claimed the biocompatibility of this nanoprobe toward both HL-60 and HepG2 compared to doxorubicin (Dox). However, the applied concentration of the nanoprobe was more than 1000 times lower than Dox. Furthermore, as no selectivity index was calculated, therefore, it would not be wise to consider a KH1C12-aptamer-based nanoprobe as a biocompatible probe, based on the current results [71]. In another study, AuNPs-coated magnetic Fe3O4 nanoparticles (AuIONPs) were used to immobilize thiolated sgc8c aptamer (Apt-AuIONPs). Then, ethidium bromide (EB) was added to intercalate into the stem of the aptamer hairpin. When human acute lymphoblastic leukemia (ALL) CCRF-CEM cells containing solution were placed onto the Apt-AuIONPs, the CCRF-CEM cells disrupted the hairpin structure of the aptamer to release EB, resulting in a decrease in the electrochemical signal. The read-out signal could be amplified by immobilizing nitrogen-doped graphene nanosheets on the electrode surface. The aptasensor reported detecting as low as 10 CCRF-CEM cells/mL under optimal conditions. This aptasensor selectively diagnosed leukemia from control (Romas) cells, a type of human Burkitt’s lymphoma cells [73]. Moreover, this aptasensor could diagnose 10 leukemia cells at 1000 times diluted condition, compared to Haghighi et al.’s (2020) developed aptamer [71,73]. The selectivity of sgc8c aptamer toward CCRF-CEM cells was also confirmed by Shan et al. (2014) [75]. AuNPs modified with aminophenyl boronic acid (APBA-AuNPs) were used to immobilize the aptamer, and APBA-AuNPs were also reported to bind with the CCRF-CEM cell membrane. Then, the application of silver metal caused signal enhancement, and a change in signal could be detected by both quartz-crystal microbalance (QCM) and fluorescence microscopy. Like the previous study, this nanoprobe selectively diagnosed CCRF-CEM cells from the Romas cells [75]. However, the detection limit of 1160 cells/mL indicated that Khoshfetrat et al. (2017) developed a nanoprobe that was more effective in the diagnosis of a small proportion of leukemia cells in the test sample (Table 1) [73,75].

### 3.2. Detection of Cancer-Associated Biomarkers

#### 3.2.1. Surface-Antigen Detection

Accurate and sensitive detection of ALL antigen (CD10) plays a crucial role in determining the diagnosis and prognosis of hematopoietic tumors (e.g., ALL and follicle center-cell lymphoma) and other malignancies. AuNPs-based label-free immunosensor containing Ab2 antibody detects CD10 by QCM. It should be noted that a higher frequency shift was seen when the CD10 was present alone or in antigen mixture (CD10, CD19, and CD20). This technique could be used in the quantitative detection of CD10 at a concentration range of 1.0 × 10^−8^ to 1.0 × 10^−11^ M (Table 1) [76]. Similarly, MacLaughli et al. (2012) [79] developed AuNPs conjugated with polyethylene glycol (PEG) and CD molecules specific monoclonal antibodies to diagnose chronic lymphocytic leukemia (CLL). CD20 expressed CLL cells were diagnosed using surface-enhanced Raman spectroscopy and dark-field microscopy. This technique could also be used to detect cell surface CD19, CD45, and CD5 after conjugating with their corresponding antibodies [79]. It should be noted that MacLaughli et al. (2012) and Yan et al. (2015) did not conduct any experimentation using leukemia cells or clinical samples. An analysis using a clinical sample could confirm the sensitivity and reliability of their detection procedure [76,79]. Similarly, Nguyen et al. (2010) [80] synthesized surface-enhanced Raman-scattering (SERS) AuNPs for selective targeting and diagnosis of CLL. SERS AuNPs were grafted with PEG to prevent aggregation and conjugated with anti-CD19 antibodies (CD19-Ab) for selective targeting of CD19, an overexpressed marker in CCL patients. Raman spectroscopy and dark-field microscopy confirmed that the CD19-Ab functionalized SERS AuNPs diagnosed Giemsa-stained CCL cells (isolated from CLL patients) with reliable accuracy. While staining with anti-CD4, antibody-coated SERS AuNPs neither visualized CCL cells in the dark field nor showed any signs of Raman-fingerprint spectra (Figure 4) [80].

A similar antibody-based overexpressing marker targeting strategy has been reported in lymphoma diagnosis. AuNPs conjugated with an ACT-1 antibody (AuNPs-ACT-1), which interacted with CD25, a lymphoma biomarker. AuNPs-ACT-1 enhanced the visualization of CD25, expressing Karpas 299 lymphoma cells at 750 nm laser in a multi-photon microscopy system. It is noted that no CD25 negative cell line was used in this study to validate experimental outcomes or to observe antibody cross-reactivity [83]. Moghiseh et al. (2018) [84] synthesized rituximab (R-Ab)-conjugated AuNPs (R-Ab-AuNPs) for selective detection of lymphoma using spectral-photon-counting computed tomography (SPCT). R-Ab is a well-known monoclonal antibody that selectively binds with CD20-expressed Raji (lymphoma cancer) cells. SPCT images confirmed that R-Ab-AuNPs-treated Raji cells had a higher volume and brighter hue compared to controls (water, gold chlorides, or trastuzumab-AuNPs), reflecting the presence of AuNPs at higher concentrations. This strategy also displayed similar results in detecting Herceptin 2-positive SKBR3 breast-cancer cells using the trastuzumab (Herceptin 2-targeting antibody) [84]. Studies using clinical samples could give a precise idea about the diagnostic outcomes.

#### 3.2.2. Detection of Receptor Overexpression

Folic acid (FA) receptor α overexpression occurs in different solid malignancies, for example, breast, ovary, uterus, cervix, colon, kidney, brain, and testicular cancers, while β receptor overexpresses in human leukemia cells [85]. Ge et al. (2014) [81] synthesized FA-conjugated porous palladium-gold nanoparticles Pd@AuNPs (FA-Pd@AuNPs) for detection of FA-receptor overexpressed cancer cells. Pd@AuNPs mimic natural peroxidase activity. In the presence of H_2_O_2_, FA-Pd@AuNPs produced hydroxyl radicals (•OH), which interact with 3,3′,5,5′-tetramethylbenzidine to form a blue-color product. Experimentation using K562 (chronic myeloid leukemia), A549 (non-small cell lung cancer), MCF-7 (breast cancer cells), H9c2 (cardiac myocytes), and normal cells confirmed that FA-Pd@AuNPs showed excellent sensitivity and selectivity in the detection of FA antibody) [84]. Studies using clinical samples could give a precise idea about the diagnostic outcomes.

#### 3.2.3. Detection of Receptor Overexpression

Folic acid (FA) receptor α overexpression occurs in different solid malignancies, for example, breast, ovary, uterus, cervix, colon, kidney, brain, and testicular cancers, while β receptor overexpresses in human leukemia cells [85]. Ge et al. (2014) [81] synthesized FA-conjugated porous palladium-gold nanoparticles Pd@AuNPs (FA-Pd@AuNPs) for detection of FA-receptor overexpressed cancer cells. Pd@AuNPs mimic natural peroxidase activity. In the presence of H_2_O_2_, FA-Pd@AuNPs produced hydroxyl radicals (•OH), which interact with 3,3′,5,5′-tetramethylbenzidine to form a blue-color product. Experimentation using K562 (chronic myeloid leukemia), A549 (non-small-cell lung cancer), MCF-7 (breast-cancer cells), H9c2 (cardiac myocytes), and normal cells confirmed that FA-Pd@AuNPs showed excellent sensitivity and selectivity in the detection of FA overexpressed K652 and A549 cells. Detection limit for K652 cells was 28 cells/mL (Figure 5a) [81]. Similarly, Ai et al. (2012) [77] prepared fluorescein isothiocyanate (FITC)-labeled FA-conjugated AuNPs (FITC-FA-AuNPs) for targeting FA receptors in cancer cells. FITC functions as a fluorescent reporter, and FITC-containing cells emit a green fluorescence signal. Confocal-microscopy images confirmed that FITC-FA-AuNPs were uptaken by both HeLa cells (human epithelial cervical cancer) and CERF-CEM cells (Figure 5b). Additionally, the MTT assay showed that FITC-FA-AuNPs were biocompatible with HeLa cells. Experimentation using a normal cell line could confirm the selectivity and reliability of the FITC-FA-AuNPs in leukemia detection [77].

#### 3.2.4. Nucleic-Acids Biomarker Detection

Acute myeloid leukemia (AML) cells express Wilm’s tumor (WT1) gene. However, WT1 levels in plasma are extremely low. AML patients carrying MRD release a low amount of WT1 [24]. Mehn et al. (2014) [24] developed gold-coated maghemite (γ-Fe_2_O_3_) nanoparticles functionalized with thiolated single-stranded DNA (ssDNA), complementary to the WT1 sequence, which act as Raman signal amplifiers. The ssDNA-functionalized nanoparticles detected DNA containing a WT1 sequence (XGGGCGTGTGACCGTAGCTTTAACC CTGATTGCGAATAGCG, where X = Amino C6 labeled with malachite green) using surface-enhanced Raman spectroscopy (Figure 6a). The major limitations of this approach were no clinical sample was examined, and the clinical sample will need to be labeled with Raman reporter-dye molecules [24]. Likewise, Ensafi et al. (2011) [72] constructed a porphobilinogen deaminase (PBGD) gene-detection probe. The PBGD gene is highly associated with CLL. Self-assembled AuNPs were conjugated with a thiol group (–SH)-modified DNA to prepare a PBGD probe that could hybridize with the complementary segment of the target DNA. This electrochemical probe showed reproducible and reliable results (at about 40 °C) in the detection of the target oligonucleotides sequence in human serum, by electrochemical impedance spectroscopy (EIS) (Figure 6b). The detection limit of the complementary oligonucleotides-sequence concentrations was 1.0 × 10^−12^ mol/L [72].

Fusion of the Abelson murine leukemia (ABL1) gene located on chromosome nine (9), with the breakpoint cluster region (BCR) gene located on chromosome twenty-two (22), form an aberrant chromosome called Philadelphia chromosome [86]. Mazloum-Ardakani et al. (2018) developed a DNA electrochemical biosensor using poly(catechol), graphene sheets, AuNPs, and –SH modified ssDNA to detect BCR-ABL gene fusion in ALL. Here, graphene sheets enhanced the conductivity of poly (catechol), while -SH modified ssDNA hybridized with the target DNA sequence. This biosensor could detect the target DNA using different robust methods, including EIS, differential-pulse voltammetry (DPV), chronoamperometry, and cyclic voltammetry (Table 1), with a detection limit that was a 1.0 pM target DNA strand [78]. In chronic myeloid leukemia (CML), BCR-ABL fusion (e13a2 and e14a2) is one of the most common molecular alterations [82]. A standard PCR test may provide false-negative results in diagnosing e13a2 BCR-ABL fusion in CML [18]. Cordeiro et al. (2016) [82] developed AuNPs-based gold-nano beacons (Au-Nbac), in combination with Förster resonance energy transfer-based spectral codification, to detect e13a2 and e14a2 fusion. Au-Nbac showed high specificity in the detection of e13a2 and e14a2 fusion. A major concern of this strategy is false-positive results, due to cross-reactivity of ABL- or BCR-expressed mRNAs from a healthy donor. Analysis of clinical samples could draw conclusive results regarding the sensitivity and specificity of this technique [82]. Previously, Lin et al. (2010) used a thiolated hairpin-locked nucleic acids (LNA)-probe immobilized on the gold (Au) electrode (via sulfur Au interaction) to diagnose BCL–ABL fusion in CML. The hybridization reaction on the LNA probe electrode was monitored through DPV, and this strategy showed satisfactory outcomes in the diagnosis of BCL–ABL fusion from clinical samples [87].

#### 3.2.5. Proteomic Biomarker Detection

In cancer cells, proteomic changes are a common phenomenon and have drawn attention as a cancer biomarker. Magnetite (Fe_2_O_3_) partially coated with strawberry-like AuNPs (Fe@Au), in combination with mass spectrometry, diagnosed plasma-protease C1 inhibitor and heat-shock protein HSP75 as multiple-myeloma biomarkers. The study was conducted on male multiple-myeloma patients (*n* = 5) and these biomarkers were identified, comparing the expression pattern of 53 patient-derived proteins with healthy controls (*n* = 2) [74].

## 4. Gold-Nanomaterial-Based Treatments in Hematological Malignancies

The conjugation of bio-polymers, nucleic acids, antibodies, peptides, chemotherapeutic drugs, natural-bioactive compounds, or HMs targeting ligands with GNMs enhanced the therapeutic potential of these anticancer agents against HMs. GNMs-based photothermal therapy, photodynamic therapy, radiotherapy, gene therapy, and targeted drug delivery of natural or synthetic anticancer agents not only improved therapeutic outcomes but also helped to overcome drug resistance, reduction in cytotoxicity, and side effects.

### 4.1. Photothermal Therapy

PTT is a non-invasive therapeutic tool that depends on hyperthermia to eradicate cancer cells with minimum side effects [88,89]. Cells are subjected to the 41–47 °C temperature range at hyperthermic conditions. In this condition, denaturation of protein or cell membrane causes irreversible damage to the cells [88]. Laser, ultrasound, or microwave-based heating methods can be used to achieve PTT [27]. GNMs can capture electromagnetic energy and convert it into heat via the photothermal effect, thereby making them a suitable agent for the PTT [90,91]. Due to the higher light absorption and energy-conversion capacity, smaller GNMs are usually preferred for PTT [92].

Antibody-loaded AuNPs can be used for targeting HMs-specific surface receptors and, thereby, HMs. Lapotko et al. (2005) [3] used two monoclonal antibodies (primary and secondary) for targeting K562 myeloid cells. K562 cells were treated with primary antibody followed by a secondary antibody (IgG)-loaded AuNPs (AuNPs-IgG)-cluster to select leukemia cells. After that, radiation (at 532 nm) was applied from a wide-beam single-laser pulse at 5 J/cm, which destroyed AuNPs-IgG-treated cells, while cell viability did not change in the AuNPs-treated group. This laser-activated nanothermolysis strategy could be used for leukemia-cell elimination [3]. Lapotko et al. (2006) [93] later showed that the AuNPs-based dual antibody-based leukemia targeting is also effective against patient-derived leukemia cells. ALL cells derived from patients (*n* = 3) were targeted using CD-10, CD-19, and CD-20 monoclonal antibody (MAB-1), specified for three patients. Then, treatment of secondary antibody (MAB-2)-loaded AuNPs clustered along with laser irradiation (at 532 nm, 0.6 J/cm^2^) for 10 nanoseconds caused 98.5%–99.9% destruction of ALL cells by photothermally produced microbubbles. On the other hand, the same treatment induced 16%–23% death of normal BM cells derived from healthy individuals (Figure 7) [93]. Longer-wavelength (650–900 nm) radiations are more compatible with normal RBCs; therefore, AuNPs that absorb longer wavelengths could be used for clinical application [3].

PEG-modified GNR fabricated with anti-CD33 antibody (anti-CD33-Ab) exhibited about 2 to 2.5-fold higher antiproliferative activity toward HL-60 cells and K562 cells, respectively, at a 500 pM concentration compared to anti-CD33 antibody alone. Interestingly, PEG-GNR displayed excellent biocompatibility against both the cell lines up to 1 nM concentration [94]. As anti-CD33-Ab-PEG-GNR displayed higher cytotoxic activity toward HL-60 cells, Liopo et al. (2011) [94] further conducted experimentation by using HL-60 cells. A single high- or low-fluence laser irradiation at 755 nm of wavelength for 45 min, into the CD33-Ab-PEG-GNR (250 pM)-treated HL-60 cells, displayed three- to four-fold more antiproliferative activity compared to the control (laser only) [94]. It should be noted that the anti-CD33 antibody already got approval for AML treatment [95]; therefore, CD33-Ab-PEG-GNR could be used to treat AML.

Yang et al. (2014) [96] loaded anti-CD138 antibody (anti-CD138-Ab) on the surface of PEG-modified hollow AuNPs (anti-CD138-Ab-PEG-AuNPs) that selectively target A20 lymphoma cells. Treatment of female BALB/c mice with anti-CD138-Ab-PEG-AuNPs followed by NIR treatment (after 24 h) significantly decreased lymphoma growth and IgG2a expression at 4 days compared to a control. An increase in IgG2a level corresponds with higher A20 cell proliferation. It should be noted that anti-CD138-Ab exhibited more selectively toward A20 cells than R-Ab, and the tunable band for hollow AuNPs was 520–950 nm. These findings indicated that anti-CD138-Ab-PEG-AuNPs could be used for PTT-based lymphoma treatment [96].

### 4.2. Photodynamic Therapy

PDT is another non-invasive therapeutic modality that damages or kills tumor cells through the use of ROS such as •OH, superoxide anion (O_2_^•−^), and singlet oxygen (^1^O_2_) generated photochemically [97,98]. Upon irradiation of light at specific wavelengths, a photosensitizer (or light-sensitizing agent) may convert endogenous O_2_ to ^1^O_2_ to induce cell necrosis or apoptosis [99]. Like PTT, PDT offers a localized treatment where the cell destruction is limited to the photosensitizer’s located region. Therefore, PDT shows better selectivity and fewer side effects than conventional chemotherapy [97]. GNMs transport the hydrophilic photosensitizer to the target cells [98,100].

In PDT, ^1^O_2_ has a very short lifetime (several microseconds) in aqueous environments [101] and about 20 nm cellular-diffusion distance [102]. Moreover, cytotoxic ^1^O_2_ has a very short radius for action compared to the size of cancer cells (>10 μm) and the intracellular distance [101]. Sgc8-Aptarmer (Apt) exhibited selectivity toward leukemia cells, as mentioned earlier, and Apt-based selective targeting could effectively deliver the ^1^O_2_. Wang et al. (2013) [103] conjugated hydrophobic chlorin e6 (Ce6) photosensitizer molecules on the surface of Apt-fabricated GNR (Ce6-Apt-GNR). In PDT, irradiation of white light reduced the viability of Ce6-Apt-GNR-targeted CCRF-CEM cells to about 74.5%, hile NIR-laser irradiation (812 nm) for 10 min decreased cell viability to around 63%, due to the PTT killing of CCRF-CEM cells. The combination of white light and NIR-laser irradiation dramatically reduced the cell viability to below 32%. Results indicated that Ce6-Apt-GNR increased the photodestruction efficiency through dual-modal PDT and PTT therapy. PDT, PTT, or PDT/PTT caused a significant reduction in cell viability compared to an untreated control. On the other hand, Ce6-Apt-GNR displayed less phototoxic (cell viability remained at nearly 90%) effects on Ramos cells (non-target) [103]. As Ce6 shows low toxicity, it could be used as a photosensitizer in HMs treatment. Philchenkov et al. (2014) [104] later reported that Ce6 conjugation with AuNPs (Ce6-AuNPs) had no photo-enhancement effect toward Jurkat and Jurkat/A4 (subline of Jurkat cells with a multidrug-resistance phenotype) compared to free Ce6. Laser radiation at 633 nm decreased apoptosis by about 10% in Ce6-AuNPs-treated Jurkat/A4 cells compared to free Ce6, while Ce6-AuNPs acted as a potent free Ce6 against Jurkat cells (Figure 6) [104]. These findings prove that Apt-based targeting is crucial in Ce6 delivery to leukemia cells.

An intermediate of the heme biosynthesis pathway, 5-aminolevulinic acid (5-ALA) is an FDA-approved photosensitizing precursor for PDT in actinic keratosis treatment and as an imaging agent to visualize cancer cells during glioma surgery [105]. Zhan et al. (2015) [106] grafted 5-ALA with AuNPs (5-ALA-AuNPs) via electrostatic bonding for PDT. These 5-ALA-AuNPs enhanced ^1^O_2_ generation and its efficiency depends on the light sources. The viability of K562 cells decreased significantly compared to ALA alone in all three light sources: a 502 nm light-emitting diodes (LEDs), xenon lamp, and 635 nm continuous-wavelength semiconductor laser. Among these three light sources, 5-ALA-AuNPs were highly effective at LEDs in reduction (nearly 70%) in K562 cell viability compared to xenon lamp (~50%) and a 635 nm laser (~56%). AuNPs enhanced K562 cell-killing through its delivery activity of 5-ALA, not by ^1^O_2_ generation or local-field enhancement [106].

### 4.3. Radiation Therapy

Ionizing radiation causes significant lymphocytopenia when administered to target volumes [107]. To destroy cancer cells, fractionated focal irradiation is often given to cancer patients either singly or in combination with other therapies [33,108]. Local radiation decreased all lymphocyte subsets in the systemic circulation, especially the B-cell and naive T-cell population [107]. PEG-modified AuNPs (AuNPs-PEG) were uptaken by leukemic HL-60 II (p53 null) and Jurkat cells to a similar extent, although there was no targeting moiety. Interestingly, AuNPs-PEG (15 nm) with 5 Gy external radiation showed a higher sensitivity enhancement ratio against HL-60 II (1.33) compared to Jurkat cells (1.18). While AuNPs-PEG and radiation had no synergistic effects on the Jurkat cells, the combination therapy decreased 3.9% of the s-phase in the HL-60 II cell population compared to radiation alone, indicating combination therapy reduced actively replicating cell percentages. Additionally, these synergetic effects significantly changed the metabolic activity of HL-60 II cells than radiation alone [108].

### 4.4. Gene Therapy

Gene therapy is another cancer therapeutics approach, where the gene responsible for cancer development modulates to become silent [109]. The use of miRNA or siRNA combination in targeting oncogenes could offer dual modulation and inhibition of oncogenes within the same signaling pathway. Furthermore, a combination of many siRNA sequences to modulate cancer may increase therapeutic efficacy [110,111]. AuNPs have become attractive vehicles for carrying gene-silencing moieties, either alone or in combination with other anticancer drugs [109].

Tyrosine-kinase inhibitors (TKIs) control downstream signaling involved in differentiation arrest, enhanced proliferation, and death resistance [112]. However, CML patients (over 30%) required an alternative of TKIs to avoid drug resistance, side effects, or tolerance [113]. Additionally, TKI-based treatment could not eradicate the oncogenic events and etiological cause of CML. The residual BCR/ABL-positive cells remain as “oncogenic-quiescent”, which causes cancer relapse [114]. A recent study reported that PEG-modified AuNPs grafted with e14a2 antisense hairpin ssDNA oligonucleotide (AuNP-PEG-e14a2) displayed imatinib (IM) equivalent cytotoxic potential toward K562 (positive for BCR-ABL1 e13a2 transcript), while both of them had no cytotoxic effect against THP1 (negative for BCR-ABL1 e13a2 transcript). AuNP-PEG-e14a2 and IM combination treatment were 23% more effective in reducing cell viability compared to IM alone [113]. AuNP-PEG-e14a2 also caused a more than two-fold change of Bax/Bcl2 ratio, thereby shifting cells from an anti-apoptotic setting to a pro-apoptotic one. Furthermore, this fold changing (of Bax/Bcl2 ratio) was higher than the IM-treated group as well. Like IM, AuNP-PEG-e14a2 significantly increased caspase-3 expression at 48 h (Figure 8). This gene-silencing strategy is in line with European LeukemiaNet’s recent recommendations. It should be mentioned that AuNP-PEG-e14a2 was not effective against BV173 CML cells (e14a2 positive), while IM was effective against BV173 cells [113]. Therefore, an extensive in vivo study in different leukemia models should be performed before considering AuNP-PEG-e14a2 as an alternative to TKI.

Zaimy et al. (2016) [115] screened five antisense oligonucleotides (AOs) against BAG1, MDM2, Bcl-2, BIRC5 (survivin), and XIAP genes to target acute myeloid leukemia subtype 2 (AML-M2) cells (Table 2). These genes are considered anti-apoptotic genes, as their respective proteins are directly or indirectly linked with anti-apoptotic activities. AuNPs were functionalized with AOs, one anti-CD33(+)/CD34(+) aptamer, and Dox (FNGs). MTT-assay results confirmed the FNGs were equally potent to Dox. Though the addition of Dox with the FNGs did not improve overall cytotoxic outcome compared to Dox, the FNGs (300 μg/mL) were significantly effective in the downregulation of BCL2, BAG, survivin, and MDM2 expression in AML-M2 cells compared to Dox (300 μg/mL) [115].

Nucleolin (NCL) is a multifunctional protein highly expressed in the nucleolus. NCL regulates mRNA stability and translation of some genes responsible for tumor progression. AS1411 is a DNA aptamer (26-nucleotide) that specifically interacts with the external NCL domain and has reached phase II clinical development in cancer treatment [116]. On the other hand, anti-221 is an AO that suppresses miR-221, which is linked with cancer progression [117,118]. Deng et al. (2018) [119] functionalized AuNPs with a novel nuclear-localization-signal (NLS) peptide, AS1411, and anti-221 (NLS-AS1411-anti-221-AuNPs) that targets NCL, miR-221, nuclear-factor kappa-light-chain-enhancer of activated B-cells (NFkB), and DNA methyltransferase 1 (DNMT1)-positive AML cell lines. Via in vitro experimentation using NLS-AS1411-anti-221-AuNPs against Jurkat, Kasumi-1, K562, HL60, NB4, Thp1, Molt4, C1498, 293T, and U937 cells, Deng et al. (2018) [119] confirmed that NCL-miR-221-NFkB-DNMT1 signaling is involved in AML. Furthermore, intraperitoneal injection of NLS-AS1411-anti-221-AuNPs (100 μL/mouse) into the BM of C1498 leukemia cells bearing C57BL/6 mice significantly suppressed the expression of miR-221 and DNMT1, while it increased p15INK4B and p27kip1 levels. Moreover, the nanoconjugate extended overall survival rate, decreased WBC count, reversed splenomegaly, inhibited blasts in BM, and lung metastasis in this preclinical AML-induced animal model. Finally, the excellent biocompatibility of NLS-AS1411-anti-221-AuNPs indicated that the nanoconjugate could be used in combination with an NFkB inhibitor (e.g., Bay-11) to attain an optimal anti-leukemic effect with minimal adverse effects [119]. Later, Deng el al. (2019) [120] conjugated FA, AS1411, anti-221, and Dox with AuNPs (FA-AS1411-anti-221-Dox-AuNPs) that target the miR-221 network as well as P-gp (Figure 9). FA-AS1411-anti-221-Dox-AuNPs displayed antiproliferative activity toward drug-resistant K562 cells and three AML-patient-derived leukemia cells, along with improved selectivity. Therefore, this multifunctional nanoparticle could overcome multidrug resistance in leukemia [120].

### 4.5. Improved Delivery of Chemotherapeutic Drugs, Peptides, Antibodies, or Bioactive Compounds

#### 4.5.1. Delivery of Conventional Drugs

Chemotherapy is widely used in leukemia treatment. Poor bioavailability, shorter half-life, drug resistance, and side effects are the major limitations of chemotherapeutic drugs [121,122,123,124,125,126]. Conjugation of chemotherapeutic drugs (Figure 10) with GNMs improved selectivity, efficacy, and potency against HMs.

Dox is an anthracycline group of antibiotics used in cancer chemotherapy [127]. To improve Dox selectivity and efficacy toward leukemia cells, Dizman et al. (2021) [128] conjugated Dox with AuNPs (Dox-AuNPs). MTT-assay results showed that Dox-AuNPs (5 μL) were more potent compared to Dox (5 μL) against K562 and HL-60 cells. However, at the same concentration, Dox-AuNPs were about 1.5-fold less cytotoxic compared to Dox, indicating the selectivity of Dox-AuNPs toward leukemia cells [128]. Daunorubicin (Dau) is another anthracycline group of antibiotics having Dox-like antileukemic effects [127,129]. Clinical applications of Dau have been decreased due to toxicities such as nausea, vomiting, diarrhea, infections, dental ulcer, lower ejection fraction, and myocardial ischemia [121]. To reduce the adverse effects and increase efficacy, Danesh et al. (2015) [130] loaded Dau into the surface of sgc8c aptamer (Apt) conjugated AuNPs (Apt-Dau-AuNPs). As mentioned earlier, the Apt selectively targeted ALL cells, and confocal-microscopy results also showed similar results for Apt-Dau-AuNPs. MTT-assay results revealed that Apt-Dau-AuNPs were about 14% more potent in reducing Molt-4 cells (human ALL T-cell, target) than free Dau (0.5 µM). On the other hand, free Dau exhibited nearly 18% higher antiproliferative activity against U266 (B lymphocyte human myeloma, non-target) cells compared to the Apt-Dau-AuNPs. It should be noted that this nanoconjugate released four-fold more Dau at pH 5.5 compared to normal blood pH (7.4). This drug-delivery system could be used in controlled and targeted delivery of Dau into human ALL cells [130].

Fludarabine Phosphate (FLP) is a commercially available chemotherapeutic drug used in the treatment of HMs. FLP interferes with DNA synthesis in HMs [122,124]. However, FLP has many side effects such as lymphopenia, severe autoimmune hemolytic anemia, gastrointestinal toxicity (nausea, vomiting), hair loss, and more [122,123,124]. To improve the selectivity and efficacy of FLP, Song et al. [124] loaded FLU on FA-conjugated AuNPs (FLP-FA-AuNPs). MTT-assay results showed that FLP-FA-AuNPs (2 mM) displayed 36% to 48% more cytotoxic activity toward KG1 cells (a type of AML cells) compared to FLP alone (2 mM), after 24 h to 48 h. Interestingly, AuNPs alone had no cytotoxic activity at the test concentration [124].

Phosphatidylinositol 3-kinase (PI3K) and protein kinase B (Akt), the mammalian targets of the rapamycin (mTOR)-signaling pathway (PI3K-Akt-mTOR), are dysregulated in HMs [131]. Rapamycin (RAP) is a bacterial macrolide, having anticancer and immunosuppressant properties. RAP targets mTOR, which controls cell division, protein biosynthesis, and cell-cycle progression (G1-S phase) [131,132]. T cell immunoglobulin and mucin domain 3 (Tim-3) receptor overexpressed in AML. Tim-3 is involved in the trafficking of galectin-9 protein that protects AML cells against the host immune system [26]. To improve RAP delivery in AML cells, Yasinska et al. (2017) [26] loaded RAP on the surface of anti-Tim-3 single-chain antibody (anti-Tim-3-ScAb)-conjugated AuNPs. In THP-1 human AML cells, the nanoconjugates significantly downregulated mTOR-mediated phosphorylation of eukaryotic initiation factor 4E binding protein (eIF4E-BP) [26]. Phosphorylated eIF4E-BP is overexpressed in HMs and is involved in cancer progression [133]. Furthermore, the nanoconjugate completely abrogated mTOR activity at a 50-times lower concentration compared to free RAP [26].

In AML, an initial mutation in nucleophosmin 1, isocitrate dehydrogenase 2, tet methylcytosine dioxygenase 2, or DNA methyltransferase 3 alpha genes provide the survival advantage of the hematopoietic stem cells. After that, a secondary mutation is linked with the actual malignant transformation of the myeloid cell [134]. Among the secondary mutations, FMS-like tyrosine kinase 3 (FLT3) or FLT3 with an internal tandem duplication (ITD) i.e., FLT3-ITD is reported in about 25% to 30% of the cases of all AML [134,135]. cBioPortal results also showed higher frequencies of the listed genes in leukemia (Figure 2b). TKI, for example, midostaurin (MDS), sorafenib (SOR), lestaurtinib (LES), and quizartinib (QUI), are widely used to suppress FLT3 kinase [134,136]. Simon et al. (2015) [136] loaded MDS into the surface of pluronic-F127-modified AuNPs (MDS-PI-AuNPs) to improve the efficacy of MDS. Dark-field microscopy images showed that MDS-PI-AuNPs internalized into OCI-AML3 and THP1 cells. Drug-release test results confirmed that more than 56% of MDS released from MDS-PI-AuNPs under the simulated cancer-cell condition, which resulted in a significant reduction in the viability of OCI-AML3 and THP1 cells compared to free MDS. Similarly, to improve the efficacy and delivery of sorafenib, lestaurtinib, and quizartinib, Petrushev et al. (2016) [134] loaded SOR on pluronic-coated AuNPs, and gelatin-coated AuNPs were conjugated with LES or QUI. Dark-field microscopy or TEM images revealed that the nanoconjugates internalized into THP1 and OCI-AML3 cells. At the same time, cell counting or MTT assay assured that these nanoconjugates were more effective in reducing THP1 and OCI-AML3 cells viability compared to their corresponding TKIs alone. Furthermore, the nanoconjugates induced significantly higher apoptosis and FLT3 protein expression than their respective TKIs alone [134]. Simon et al. (2015) and Petrushev et al. (2016) did not conduct experimentation using normal cells. Therefore, the selectivity of the nanoconjugates could not be calculated and compared with free TKIs [134,136].

In CML, the BCR–ABL1 fusion gene encodes a tyrosine kinase, and its continuous expression leads to uncontrolled cell proliferation [86]. TKI, IM is used as a first-line treatment for CML with BCR–ABL1 gene fusions [137]. Ganbold et al. (2013) [138] conjugated IM, topotecan (Topo), and 4-carboxylic benzoic acid (CBT) on the surface of AuNPs. Topo is a DNA topoisomerase I inhibitor that acts as a fluorescent-reporter molecule, while CBT targets transferrin protein. Raman spectroscopy and dark-field microscopy studies confirmed that the nanoconjugate entered into the K562 cells via the receptor-mediated endocytosis and released the drugs (IM and Topo) after glutathione (2 mM) treatment [138].

Then, 6-Mercaptopurine (6-MTP) is commonly used to treat human leukemias and some other diseases, including systemic lupus erythematosus, inflammatory bowel disease, and rheumatoid arthritis [139,140]. Short plasma half-life and repaid renal clearance decrease the therapeutic effects of 6-MTP. Podsiadlo et al. (2008) [141] conjugated 6-MTP with AuNPs (6-MTP-AuNPs) to improve its delivery and thereby efficacy. Such 6-MTP-AuNPs exhibited higher cytotoxic activity toward K562 cells compared to 6-MTP alone, while AuNPs had no antiproliferative activity at the same concentration of 6-MTP-AuNPs or 6-MTP. Higher antiproliferative activity of 6-MTP-AuNPs could be linked with their positive surface charge (+19 mV) that facilities crossing the negatively charged plasma membrane, where 6-MTP is neutral or slightly negative. However, 6-MTP-AuNPs displayed 6-MTP equivalent activities in the induction of apoptosis or necrosis in K562 cells. Therefore, it was hypothesized that 6-MTP-AuNPs will show a 6-MTP-like mechanism of action(s) [141].

Methotrexate (MTX) is a widely used chemotherapeutic drug used to control cell division in HMs [142]. Cytopenias, organ toxicity (liver, kidney, and skin), and mucositis are the major side effects of MTX [143] To improve the selectivity of MTX, Egusa et al. (2014) [144] conjugated with AuNPs (MTX-AuNPs). MTX-AuNPs exhibited higher antiproliferative activities toward TPH-1 cells compared to MTX alone at the same concentration (1 nM, 2 nM, or 5 nM), while MTX-AuNPs displayed MTX-equivalent cytotoxicity against normal hematopoietic stem/progenitor cells, indicated that AuNPs conjugation enhanced the selectivity of MTX. Similarly, MTX-AuNPs showed higher therapeutic potential compared to MTX in the primary AML cell-bearing non-obese severe diabetic combined with immunodeficiency gamma (NSG) mice following intravenous injection (0.25 mg/kg, twice per week) for 6 weeks. More importantly, MTX-AuNPs displayed excellent biocompatibility and reduced anemia signs, BM, and splenic leukemia burden, compared to MTX alone. These results indicated that AuNPs conjugation enhanced the MTX therapeutic index [144].

FA-receptor targeting efficiency of FA-conjugated nanoparticles depended on the cell-surface FA-receptor expression level [145]. Bortezomib (also called Velcade) is a 26S proteasome inhibitor widely used in the treatment of multiple myeloma [146]. Patra et al. (2008) [126] reported that conjugation of Velcade (Val) with AuNPs (Val-AuNPs) significantly reduced its apoptotic activity in multiple-myeloma cells (RPMI and U266). On the other hand, functionalization of Val-AuNPs with FA retained the free Val equivalent apoptotic activity in the RPMI and U266 cells. RPMI and U266 cells might express normal cells or plasma cells such as FA receptors, therefore, an increase in apoptotic activities was not observed after Val-FA-AuNPs treatment [126,145,147]. In the FA-receptor overexpressed solid-cancer cells, Val-FA-loaded nanoparticles exhibited higher anticancer activities [146].

Arsenic trioxide (As_2_O_3_) is a traditional Chinese medicine, known for its anti-cancer effects. Thus, As_2_O_3_ has drawn considerable attention in leukemia treatment. Guo et al. (2009) [125] investigated the antileukemic effects of As_2_O_3_- and As_2_O_3_-conjugated 3-mercaptopropionic acid (MPA)-capped AuNPs (As_2_O_3_-MPA-AuNPs) against drug-sensitive leukemia K562 cells and adriamycin-resistant K562/A02 (KA) cells. MTT-assay results showed that MPA-AuNPs conjugation significantly enhanced antiproliferative activity of As_2_O_3_ toward K562 and KA cells. AuNPs increased permeability of K562 and KA cells, thereby facilitating As_2_O_3_ uptake into the cancer cells. These in vitro results indicated that AuNPs could inhibit P-gp function, a protein responsible for drug efflux from cancer cells [125].

#### 4.5.2. Improve Delivery of Antibody Drugs

AuNPs fabricated with polyclonal anti-myeloma antibody (AbMM) decreased the viability of SP2OR multiple-myeloma cells in a concentration-dependent manner (5–25 μg/mL). AbMM-AuNPs exhibited about two-fold higher anticancer activity toward SP2OR cells compared to AbMM alone. AbMM-AuNPs also arrested the SP2OR cells at the G2/M phase or S phase in the mice model. Furthermore, AbMM-AuNPs induced apoptosis by increasing the expression of cyclin-dependent kinase inhibitors (p21 and p27). These results indicated that AbMM-AuNPs could be used to inhibit multiple myeloma growth [148], while Bhattacharya et al. (2007) [149] reported that naked AuNPs exhibited remarkable antiproliferative activity toward OPM-1, RPMI-8266, and U-266 myeloma cells. The AuNPs (20 μg) arrested OPM-1, RPMI-8266, and U-266 cells at the G1 phase, by increasing p21 and p27 expression [149]. These findings indicated that AuNPs alone could have anti-myeloma activity.

Despite the therapeutic advancement in the treatment of CLL, progression to the Richter syndrome or cancer relapse remains an undruggable issue [150]. R-Ab already got FDA approval in the treatment of lymphoma [151]. To overcome the current therapeutic limitations of R-Ab, BOCA et al. (2016) [150] fabricated AuNPs with R-Ab, a monoclonal antibody that selectively targets CD20-expressed lymphoma or CLL cells and induces cell death [150,152]. Dark-field microscopy and TEM results confirmed that the AuNPs-R-Ab nanoconjugate was internalized in both the CLL cells (fibroblast-like HS 505.T cells and malignant lymphocytes CLL-AAT). MTT and cell counting assays showed that AuNPs-R-Ab was more potent in reducing cell number compared to R-Ab. Moreover, AuNPs-R-Ab was significantly effective in apoptosis induction as well as suppressed the expression of membrane-spanning 4-domains, subfamily A, member 1 (MS4A1) gene, or protein in CLL-AAT and HS 505.T cells compared to R-Ab alone. Furthermore, a confocal-microscopy study showed that such changes in MS4A1 expression resulted in a significant decrease in CD20 expression [150]. Similarly, García BE et al. (2014) [153] prepared R-Ab-conjugated AuNPs (R-Ab-AuNPs) that target Raji Burkitt’s lymphoma cells. R-Ab-AuNPs significantly downregulated anti-apoptotic Bcl-2 family proteins compared to a control, indicating that R-Ab-AuNPs could induce apoptosis. A flow-cytometry-based apoptotic-cell analysis should be performed to confirm the apoptotic activity of R-Ab-AuNPs [153]. Previously, the selectivity of R-Ab toward the lymphoma cells was also reported by Weiss et al. (2009) [154]. R-Ab conjugated on the surface of PEG-modified AuNPs exhibited selective binding and internalization into the Z138 human mantle-cell lymphoma cells (CD20 positive) but had no such activity in the SKBR3 breast-cancer cells (CD20 negative) [154].

#### 4.5.3. Peptide-Based Anti-Angiogenic Therapy

CML cells with BCR–ABL1 gene fusions trigger many cells and molecular events in BM, which are linked with the disease prognosis and progression [155]. In the BM-tumor microenvironment of CML patients, paracrine and autocrine communication between BM cells and the malignant cells are crucial for the evolution and modulation of the niche [155,156]. CML cells secrete exosomes that induce new blood-vessel formation in the BM. Roma-Rodrigues et al. (2019) [86] synthesized antiangiogenic peptide (AP)-functionalized oligo ethylene glycol (OEG)-coated AuNPs (AP-OEG-AuNPs) that impaired K562-induced new blood-vessel creation in the chorioallantoic-membrane model (Figure 11). K562 exosomes (50 μg/mL) did not affect IL8 and VEGFA expression, but they increased VEGFR1 by 200-fold. AP-OEG-AuNPs (16.4 nM) blocked the activity of K562 exosomes by silencing the VEGFR1-mediated signaling pathways [86]. The exosome-based effects of CML on the molecular pathways could be used to invent potential therapeutic strategies to regulate the patient BM tumor microenvironment.

#### 4.5.4. Bioactive Compounds

Biogenic AuNPs (b-AuNPs) synthesized from Boswellia serrata or Lens culinaris displayed cytotoxicity toward Human HL-60/vcr, Murine C1498, and 32D-FLT3-ITD cells, which was comparable to the mitoxantrone. However, b-AuNPs were less cytotoxic toward HUVEC cells compared to mitoxantrone [157,158]. In a 7,12-dimethylbenz[a]anthracene-induced AML mice model, b-AuNPs (1 mg/kg body weight) such as mitoxantrone significantly increased lymphocyte, platelet, RBC parameters, and anti-inflammatory cytokines level (IFNα, IL4, IL5, IL10, and IL13), while decreasing the level of pro-inflammatory cytokines (TNFα, IFNY, IL1, IL6, IL12, and IL18) compared to untreated mice [157,158]. Furthermore, b-AuNPs and mitoxantrone significantly increased the expression of sphingosine-1-phosphate receptor-1 and sphingosine-1-phosphate receptor-5 mRNA compared to the control mice [157,158]. Ahmeda et al. (2020) [157,158] also confirmed that like mitoxantrone, b-AuNPs also significantly decreased the infiltration of leukemic myeloblasts in the spleen and liver compared to the untreated mice. Ahmeda et al. (2020) [159], Zangeneh et al. (2020) [160], and Hemmati (2020) [161] also got similar in vitro and in vivo results from b-AuNPs synthesized from *Camellia sinensis*, *Hibiscus sabdariffa* flower, and *Hymus vulgaris* leaf, respectively. Unlike previous studies, instead of mitoxantrone, Ahmeda (2020) and Zangeneh et al. (2020) used Dau as a control drug, while Hemmati (2020) used Dox as a control drug [159,160,161]. These findings indicated that b-AuNPs are more selective toward leukemia cells compared to normal cells, therefore, AuNPs could be considered as a potent therapeutic agent in leukemia treatment.

### 4.6. Reactive Oxygen Species-Mediated Cytotoxicity

Reactive oxygen species (ROS) are special types of oxygen (O_2_)-carrying reactive molecules, which play crucial roles in many cellular activities, for instance, promoting cell growth at basal level [162]. ROS plays a crucial role in hematopoiesis by regulating of differentiation, self-renewal, and the balance between quiescence and proliferation of hematopoietic stem cells (HSCs) [163]. HSCs are found in relatively hypoxic environments where anaerobic metabolism drives HIF1 and FOXO transcription to maintain quiescence and HSC self-renewal [164]. On the other hand, an elevated ROS level is highlighted as one of the key players that underlie the acquisition of the various hallmarks of cancer, including hematopoietic malignancy [165,166]. ROS, however, could enhance cytotoxic activities at an adequately high concentration, often involving cellular apoptosis or necrosis [167,168]. Vectorized GNMs act as a catalyst in ROS production [169,170]. ROS generated from vectorized GNMs could induce local damage to the cancer cells, while restraining minimum damage to neighboring cells or GNMs-free cells [171].

GNR, in conjugation with ultra-small platinum nanoparticles (USPN), had shown multienzyme-like activities, which caused fluctuation of intracellular ROS level [172]. Western blotting results confirmed that 24 h of GNR-USPN (20 pM) treatment significantly upregulated autophagic protein Beclin-1 expression in K562 cells, thereby triggered cellular autophagy. Additionally, beclin-1 downregulated fusion protein BCR-ABL expression, leading to downregulation of AKT and PI3K phosphorylation. Furthermore, the same concentration of GNR-USPN also arrested about 37% of K562 cells at the mitotic phase (G2/M), and this growth retardation had significantly driven CML apoptosis compared to an untreated control. These in vitro findings give the idea that CML cells might be vulnerable to ROS fluctuation triggered by the enzyme-like activities of GNR-USPN. A drawback of this study was all the experiments were carried out in only one cell line. Therefore, its effects on normal cells could help to understand the selectivity of GNR-USPN [172]. Minai et al. (2013) [171] tagged R-Ab to a gold nanosphere (GNHs) surface for selective targeting of BJAB cells (Burkitt lymphoma B-cells). Time-lapse microscopy studies using some fluorescent markers confirmed that irradiation of femtosecond pulses (at 550 nm) causes a significant increase in intracellular ROS in the targeted cells. While co-culture experimentation using K562 and BJAB cells showed that the nanoconjugate could not induce a noticeable ROS level in the non-targeted K562 cells. Six femtosecond pulses increased the intracellular ROS level about 27% in the BJAB cells. It should be noted that ROS-mediated damage in the targeted cancer cells was directly proportionate with the number of femtosecond pulses (Table 3) [171].

### 4.7. Induction of Apoptosis

AuNPs modified with PEG (AuNPs-PEG) exhibited antiproliferative activity against K562 cells. AuNPs-PEG gathered into cytoplasmic vacuoles and significantly decreased mitochondrial-membrane potential, resulting in apoptotic changes. Morphological characteristics of the apoptotic cells, such as fragmented nuclei, membrane blebbing, and apoptotic bodies, were observed under the light microscope after 24–72 h of treatment. Furthermore, AuNPs-PEG significantly arrested about 15%–75% of K562 cells at 24–72 h. Together the data indicated that AuNPs-PEG-induced apoptosis in the K562 cells by controlling mitochondrial intrinsic apoptotic pathway [173]. Similarly, 25 μg of anti-VEGF antibody (VF-Ab) conjugated AuNPs (VF-Ab-AuNPs) that exhibited significantly higher apoptosis (40%–60%) toward patients (*n* = 7) with derived CLL B (B-chronic lymphocytic leukemia) cells, compared to the same concentration of VF-Ab. Western blotting results showed that VF-Ab-AuNPs (25 μg) treatment resulted in PARP cleavage, while no such responses were observed after VF-Ab treatment. PARP cleavage was associated with a decrease in caspase 3, Mcl-1, and Bcl-2 expression. As no densitometry studies were conducted, the level of significance could not be calculated from the Western blotting results [174].

Verbascoside (VER) is a hydrophobic drug candidate derived from *Banchunmaxianhao,* reported to have strong anticancer activities [175,176]. Co-treatment of VER with poly (N-isopropyl acrylamide) (PNM)-modified gold nanoshells (GNSs) (PNM-GNSs) displayed synergetic activity toward drug-resistant KA leukemia cells. DNA fragmentation and apoptotic bodies were visualized under microscopic examination after VER and PNM-GNSs treatment. This co-treatment significantly increased the apoptotic activities in KA cells by modulating the caspase 3, 8, and 9 signaling pathways and reduced tumor volume in KA-tumor-bearing female nude mice. Results indicated that VER-loaded PNM-GNSs could be used in leukemia treatment [176].

It is well known that CpG oligodeoxynucleotides (CpGs) uniquely induce cytotoxicity toward lymphoma. To improve the efficacy of CpGs, Lin et al. (2020) [177] conjugated tri-ethylene-modified CpG (class B and class C) with AuNP (tmCpG-AuNP) to target lymphomas. Both classes of tmCpG-AuNPs were more potent toward RC (high-grade B lymphoma), JeKo-1 (mantle-cell lymphoma, TP53 deficient), Mino (mantle-cell lymphoma, TP53 mutated), and REC-1 (mantle-cell lymphoma, TP53 proficient) cells compared with free CpGs, while both the tmCpG-AuNPs were not toxic against dendritic cells. An annexin V-PI apoptosis assay using mouse lymphoma A20 cells confirmed that tmCpG-AuNPs were 4.6-fold more potent in inducing apoptosis compared to free CpGs. Moreover, tmCpG-AuNPs increased CD19, CD20, and CD47 protein expression on the lymphoma cells, thereby enhanced the anti-lymphoma effects. Finally, in the A20-tumor-bearing BALB/c mice, both the tmCpG-AuNPs reduced tumor volume at the injection site. However, only class B tmCpG-AuNPs significantly enhanced survival time compared to free CpGs [177].

b-AuNPs prepared from *Sargassum muticum* water extract displayed antiproliferative activity toward K562, Jurkat, HL-60, and CEM-ss cells (acute lymphocytic leukemia). MTT-assay results indicated that the b-AuNPs displayed higher antiproliferative activity toward K562 cells compared to Jurkat, HL-60, and CEM-ss cells (Table 4). The b-AuNPs induced significantly arrest K562 cells at the sub-G0/G1 and G2/M phases of the cell cycle after 12 h as well as induced early and late apoptosis after 6 h, compared to a control. Additionally, K562 cells displayed the typical features of apoptosis including cell shrinkage, membrane blebbing, chromatin condensation, and apoptotic body formation after 24–72 h of b-AuNPs (4.22 μg/mL) treatment. Furthermore, the same treatment significantly increased the expression of caspase-3 and -9 after 24 h. These findings indicated that the mitochondrial intrinsic pathway was activated to produce the apoptogenic responses [178]. Using an MTT assay, Gautam et al. (2017) reported that b-AuNPs prepared from *Ocimum sanctum* leaf extract displayed cytotoxicity toward Dalton’s lymphoma (DL) cells. b-AuNPs decreased mitochondrial-membrane potential (ΔΨm) and, thereby, induced apoptosis through the mitochondrial pathway. A DNA-fragmentation assay also confirmed that b-AuNPs (100 μg) treatment caused higher nuclear condensation and DNA fragmentation (at 50 bp, 500 bp, 750 bp, and 1000 bp) compared to the untreated control. Additionally, b-AuNPs (100 μg) significantly arrested about 8.4% of DL cells at the G0/G1 phase. A limitation of this study was only one cell line was used to validate the experimental results. Experimentation using a normal cell line could report the selectivity of the b-AuNPs [179]. Shahriari et al. (2016) [180] prepared hydroxyl-capped b-AuNPs using *Camellia sinensis* leaf extract and conjugated b-AuNPs with asparagine (Asn). MTT-assay results showed that Asn-b-AuNPs displayed concentration-dependent (3–300 μg/mL) antiproliferative activity toward CCRF-CEM cells at 39 °C. A similar concentration-dependent trend in apoptosis induction was also seen under the same condition. Asn-b-AuNPs (30 μg/mL) induced 45% apoptosis in CCRF-CEM cells compared to the untreated control. While Asn-b-AuNPs (300 μg/mL) significantly decreased expression of matrix metalloproteinase 2 (MMP-2) [180]. MMP-2 induces cell growth, invasiveness, migration, and angiogenesis in leukemia [181]. At a higher concentration (300 μg/mL), the proportion of necrotic cells also increased. Therefore, it would not be wise to use the higher concentration of this nanoconjugate. Shahriari and co-workers did not examine Asn-b-AuNPs effects on other cells, including normal cells, under the mild hyperthermic condition. Therefore, selectivity of this nanoconjugate is still an issue for biomedical applications [180].

As mentioned earlier, FA receptor β-overexpressed in human leukemia cells [85]. GNR conjugated with silica and folic acid (F-Si-GNRs) showed higher selectivity toward EL4s lymphoblastic leukemia cells compared to spermatogonial cells (SSCs) with an IC_50_ value of <75 μM and >75 μM, respectively. Furthermore, an annexin V-FITC/PI-based flow-cytometry assay confirmed that F-Si-GNRs (100 μM) significantly increased apoptosis in EL4s cells (51.1% ± 6%) compared to SSCs cells (32.9% ± 2%). Microscopic examination showed that upon internalization, F-Si-GNRs caused mitochondrial damage to active apoptosis signaling [182].

## 5. Challenges of Using GNMs in HMs from Pharmacological and Toxicological Point of View

GNMs have already reached clinical trials in solid-cancer therapy [183]. This achievement draws the attention of scientists for their usage in HMs. Though GNMs have shown promise in HMs as a theranostic agent, overcoming the following challenges and concerns could pave the way for their clinical applications in HMs.

### 5.1. Selectivity of Gold Nanomaterials

A fundamental issue of cancer drugs is their low therapeutic index and poor selectivity toward cancer cells compared to normal cells. The anticancer agents with poor selectivity destroy cancer and normal cells, thereby posing adverse effects that limit the efficacy and safety of treatment [144]. Calculation of a selectivity index helps to understand the selectivity of an anticancer agent [184]. The selectivity of some GNMs toward HMs could not be calculated from some of the reported studies, as experimentation was conducted only against the cancer cells. Therefore, the effects of the GNMs on the normal cells remain a major concern despite the promising results against HMs. Experimentations on the normal cells could give an idea about their selectivity toward HMs.

### 5.2. Preclinical Experimentation in a Complex System

Many of the reported studies related to GNMs-based treatments were performed in the 2D-cell-culture systems. This system lacks the complexity of the cancer microenvironment. Experimentation using 3D co-cultures with supporting cells (e.g., BM cells), or organ-on-a-chip technologies that mimic the in vivo system, could be considered as a better reflection of experimental results. These advanced-culture systems offer better cell-to-cell interactions and paracrine interactions with the cancer cells that, more importantly, mimic the BM microenvironment [156]. However, many important aspects, including immunomodulatory activity, vascularization, perfusions, and effects on the other major organs, are still lacking in the 3D-culture models [156,185]. Therefore, extensive in vivo studies using theranostic GNMs should be conducted to unveil the key questions such as effects on the immune system, absorption, distribution, efficacy, and safety.

### 5.3. Immunomodulatory Effects

Immunosuppression is a common phenomenon in HMs; therefore, therapeutic agents having immunostimulatory effects could offer benefit to patients with HMs [25,186,187,188,189,190]. Undesirable interactions of GMNs with plasma, red blood cells, white blood cells, or platelets may pose an immediate threat to the biodistribution, efficacy, and biocompatibility of GMNs as cancer nanomedicine [191]. Many factors contribute to the immunomodulatory effects of GMNs, including surface chemistry, composition, shape, size, protein-binding ability, individual difference, dosage, and route of administration [192,193]. The literature showed that GMNs depending on these factors displayed both immunostimulatory and immunosuppressing activities [192,193,194]. Simpson et al. (2010) reported that mixed monolayer AuNPs (<5 nm) surprisingly increased the WBC count in female BALB/c mice at concentrations higher than 30 μM [195]. As most of the studies (discussed in Section 4) reported that the therapeutic benefits of GMs in HMs were performed only in the in vitro model, their dose-dependent immunomodulatory effects in the in vivo system could not be understood from these results.

### 5.4. Safety and Biodistribution

In vitro experimentation of nanoparticle safety does not always reflect in vivo findings, as nanomaterials metabolized in one organ may have toxic effects [185,192]. Studies in the whole organism may reflect similar findings in the clinical results, as an organism is more complex than a single cell [192]. The size, surface area, number, shape, and clearance of GNMs play a crucial role in biodistribution, accumulation, and toxicity [192,196].

GNMs are rapidly cleared from the systemic circulations and deposited in all the major organs, though the highest accumulation has been reported in the liver and spleen [192,197]. Higher clearance of GNMs occurs through urine and bile [196]. Hainfeld et al. [198] reported that about 77.5% AuNPs (1.9 ± 0.1 nm) cleared from the body of CD1 male or female mice after 5 h of intravenous injection. Moreover, the acute and chronic toxicity of the GNMs depends on the route of administration, test organism, sex, surface-coating agents, dosage, and size [33,199,200]. Zhang et al. (2010) reported that AuNPs (13.5 nm) reduced splenic index, red blood-cell counts, and body weight of male ICR mice at 2200 μg/kg dose. Among the three administration routes (oral, intraperitoneal, and intravenous), the intravenous route showed the lowest toxicity [199], while Rambanapasi et al. (2016) concluded that 14 nm AuNPs did not cause any acute or chronic toxicity in male Sprague Dawley rats at 90 μg/kg dose [200]. Additionally, In vivo results confirmed that smaller (3–5 nm) and larger (50–100 nm) GNMs are less toxic compared to intermediate-size (8–37 nm) nanoparticles, while some in vitro studies reported that intermediate-size GNMs are safe as well. Furthermore, larger GNMs (>500 nm) enter into the cells through phagocytosis, while smaller nanoparticles (<100 nm) enter via receptor-mediated endocytosis [192]. More importantly, Alkilany et al. (2010) [192] concluded that the number of active gold particles varies with the size, since the reactivity of smaller (~3) GNMs is higher than larger GNMs (>~5 nm). In another study, using an MTT assay, Mateo et al. (2015) [201] reported that 30 nm, 50 nm, and 90 nm GNMs displayed size-dependent cytotoxicity toward NHDF normal-human dermal fibroblasts cells with IC_50_ values of 17.9 mg/mL, 18.0 mg/mL, and 19.3 mg/mL, respectively. Similar dose-dependent cytotoxic activity was also observed in the LDH assay results, and GNMs drove ROS production that was linked with cytotoxicity [201], while Shuka et al. (2005) [202] concluded that GNMs reduced the ROS level in RAW264.7 macrophage cells and did not elicit proinflammatory cytokines that release TNF-α and IL1-β. It was hypothesized that GNMs were uptaken via endocytosis decreased the ROS level and were nontoxic. Recently, Balfourier et al. (2020) [203] reported that GNMs (4–22 nm) are surprisingly degraded in vitro by cells, via rapid degradation of the smallest size of GNMs. Transcriptomics studies assured that lysosomes play an active role in GNMs biodissolution (Figure 12). Future studies in other cell lines and in vivo models could unveil the possible mechanism of GNMs degradation and clearance [203]. These findings consolidated that the biocompatibility, clearance, and mechanism action(s) of GNMs is still a controversial issue [33,202,203], therefore, extensive studies are needed to clarify these controversial points.

### 5.5. Adverse Effects

Among the three administration routes, GNMs administration through the oral and intraperitoneal routes showed higher adverse effects compared to the tail-vein injection [199]. Adverse effects of GNMs include damage of cytoplasmic organelle or DNA, oxidative stress, mutagenesis, and alteration of protein expression [192]. More importantly, vectorized GNMs may cause irreversible damage to the normal cells through excessive ROS generation [32]. Consequences of such changes include a decrease in body weight, spleen index, alteration of hematocrit level, and red blood-cell counts [199]. Furthermore, Peng et al. (2019) [204] reported that GNMs could damage the endothelial cells, leading to higher extravasation and intravasation of cancer cells, thereby exacerbating cancer metastasis [204]. Breaking of the endothelial barrier could enhance the metastasis of HMs and increase the risk of secondary-tumor development [205,206,207]. Therefore, to enlighten future use of GNMs in cancer treatment, the properties of the GNMs that increase endothelial leakiness need to be determined [208]. Additionally, to resolve this issue, GNMs could be coated with bioactive compounds (e.g., polyphenols) that improve endothelial dysfunction [209]. Finally, as there is no in vivo model system to investigate endothelial leakage, in vitro studies using human endothelial cell monolayers should be performed to examine endothelial permeability and vascular leakage [210].

## 6. Conclusions

GNMs illuminate the diagnosis of HMs, metastasis of HMs, and MRD, with reliable accuracy and specificity. As only a few studies were conducted using clinical samples, future experimentations using clinical samples are still in demand, while GNMs-based PTT, PDT, radiotherapy, gene therapy, and advanced drug delivery enhanced the antiproliferative activity of conventional therapeutic agents and showed promise to overcome drug resistance and minimize the side effects. These findings indicated that GNMs-based therapies could be considered as an alternative to current treatment modalities. Additionally, tuning of GNMs size, surface modification with biocompatible photosensitizers, probes, targeting ligands, and HMs specific drug or drug candidates could improve theranostic efficiency. Furthermore, nanozyme activity of GNMs in combination with another metallic nanoparticle should be explored in the future, to diagnose and treat HMs [172,211]. One of the limitations of the many reported studies is that the testing experimentations were conducted in secondary cell lines in a 2D cell-culture environment. Additionally, testing diagnostic accuracy, reliability, and sensitivity using clinical samples, while examining the therapeutic efficacy in a 3D cell-culture model or patient-derived xenograft model, could better reflect clinical outcomes [212,213,214]. Moreover, the biocompatibility and safety of GNMs remain controversial issues. To reduce to the toxic effects, size-dependent effects should be monitored following conjugation with suitable bioactive molecules, biopolymers, or other theranostic agents. Lastly, before clinical trials, extensive preclinical studies in the in vivo model should be performed to determine the therapeutic index, selectivity, organ distribution, clearance, and mechanism action(s) of GNMs.

## Figures and Tables

**Figure 2 cancers-14-03047-f002:**
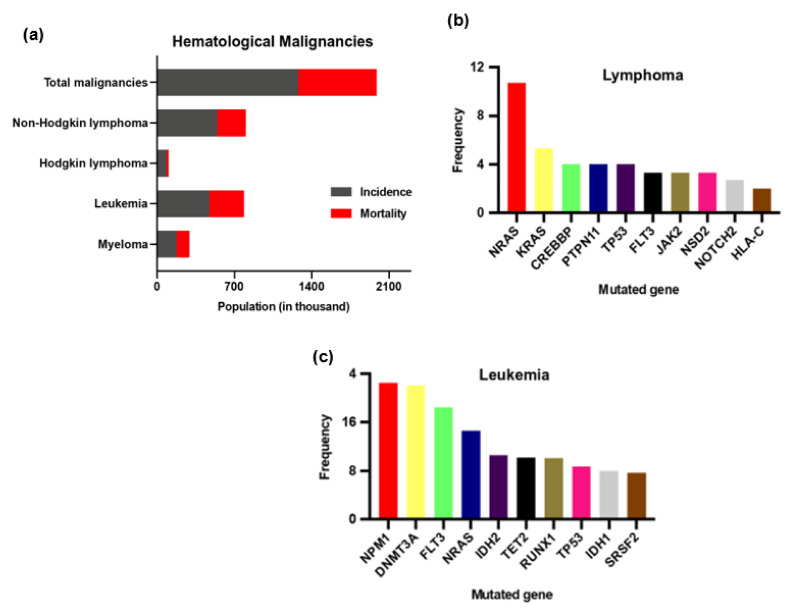
Global incidence, mortality, and frequent mutations in hematological malignancies. Overall, the incidence rate was higher compared to the death toll. The data of Figure 2 (**a**) adapted from Sung et al. (2021) [2], and their primary data source was GLOBOCAN 2020. The gene-mutation-frequency graph was prepared based on published data on (**b**) leukemia (*n* = 3768) [39] and (**c**) lymphoma (*n* = 150) [40] patients. Nucleophosmin 1: NPM1; DNA methyltransferase 3 alpha: DNMT3A; Fms-like tyrosine kinase 3: FLT3; Neuroblastoma RAS viral oncogene homolog: NRAS; Isocitrate dehydrogenase 2: IDH2; Tet methylcytosine dioxygenase 2: TET2; Runt-related transcription factor 1: RUNX1; Tumor protein 53: TP53; Isocitrate dehydrogenase 1: IDH1; Serine and arginine-rich splicing factor 2: SRSF2; Kirsten rat sarcoma 2 viral oncogene homolog: KRAS; CREB binding protein: CREBBP; Tyrosine-protein phosphatase non-receptor type 11: PTPN11; Janus kinase 2: JAK2; Nuclear receptor binding SET domain protein 2: NSD2; Notch receptor 2: NOTCH2; Major histocompatibility complex, class I, C: HLA-C. Data source: cBioPortal.

**Figure 3 cancers-14-03047-f003:**
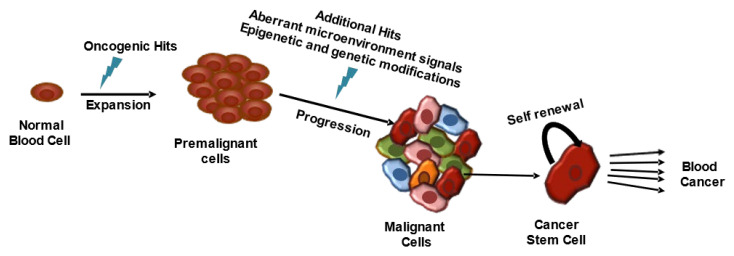
Molecular mechanism of hematological malignancies. Oncogenic hits in a normal blood cell produce premalignant cells. Additional oncogenic hits promote the development of malignant cells. Among the malignant cells, cancer stem cells have self-renewal capacity.

**Figure 4 cancers-14-03047-f004:**
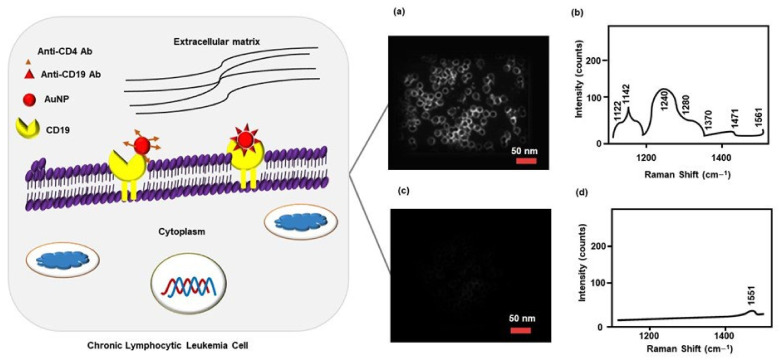
Detection of chronic lymphocytic leukemia using gold nanoparticles. Dark-field images (**a**) and accompanying Raman spectra (**b**) of Giemsa and anti-CD19 Ab-conjugated AuNPs stained CLL cells. Giemsa-stained CLL cells were not visualized after anti-CD4 Ab containing AuNPs treatment in the dark-field (**c**), and no peak was detected by Raman spectroscopy (**d**). These images indicate possible results, but do not reflect any actual experiments. Chronic lymphocytic leukemia: CCL; Cluster of differentiation: CD; Antibody: Ab; Gold nanoparticles: AuPs.

**Figure 5 cancers-14-03047-f005:**
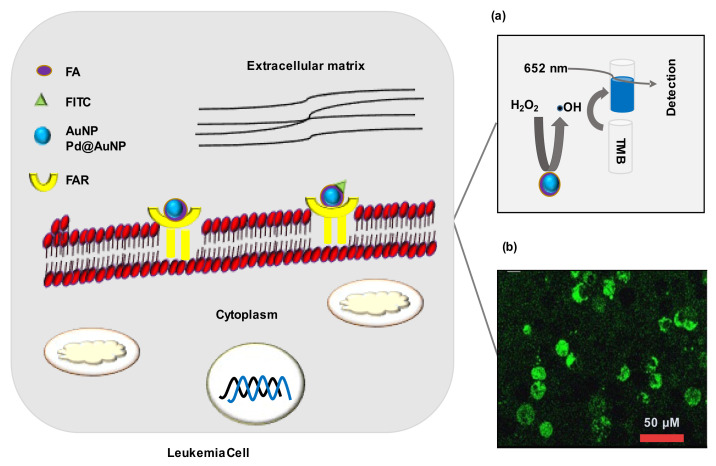
Gold-nanomaterials-based leukemia detection. FA conjugated GNMs interact with the FAR overexpressed leukemia cells. (**a**) Interaction between FAR with FA-Pd@AuNP can be confirmed by the colorimetric method. Enzymatic activity of FA-Pd@AuNP covert H_2_O_2_ to •OH. •OH interacts with TMB to form a blue color that can be detected at the 652 nm wavelength. (**b**) FA-loaded AuNPs labeled with FITC binds with FAR overexpressed leukemia cells; the interaction could be confirmed by confocal microscopy, based on the green fluorescence of the FITC reporter. These images indicate potential results, but they do not reflect any actual experiments. Gold nanomaterials: GNMs; Folic acid: FA; Folic-acid receptor: FAR; Palladium gold nanoparticles: Pd@AuNPs; Gold nanoparticles: AuNPs; Fluorescein isothiocyanate: FITC.

**Figure 6 cancers-14-03047-f006:**
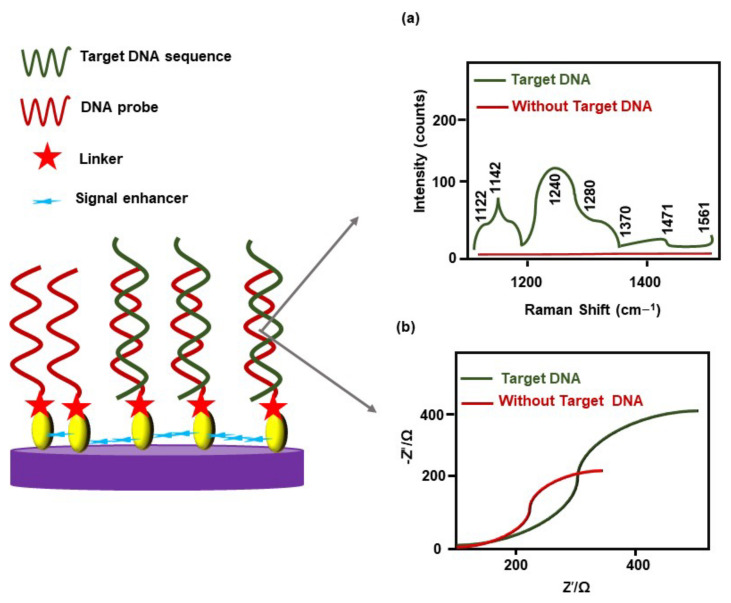
Detection of genetic changes in leukemia using AuNPs. Interaction with the target DNA sequence with the probe DNA cause changes in Raman spectra (**a**) and electrochemical signal (**b**). Signal-enhancer molecules accelerate signal changes upon binding of target nucleic-acid moiety, thereby making changes more visible and distinguishable. These images indicate potential results, but they do not reflect any actual experiment.

**Figure 7 cancers-14-03047-f007:**
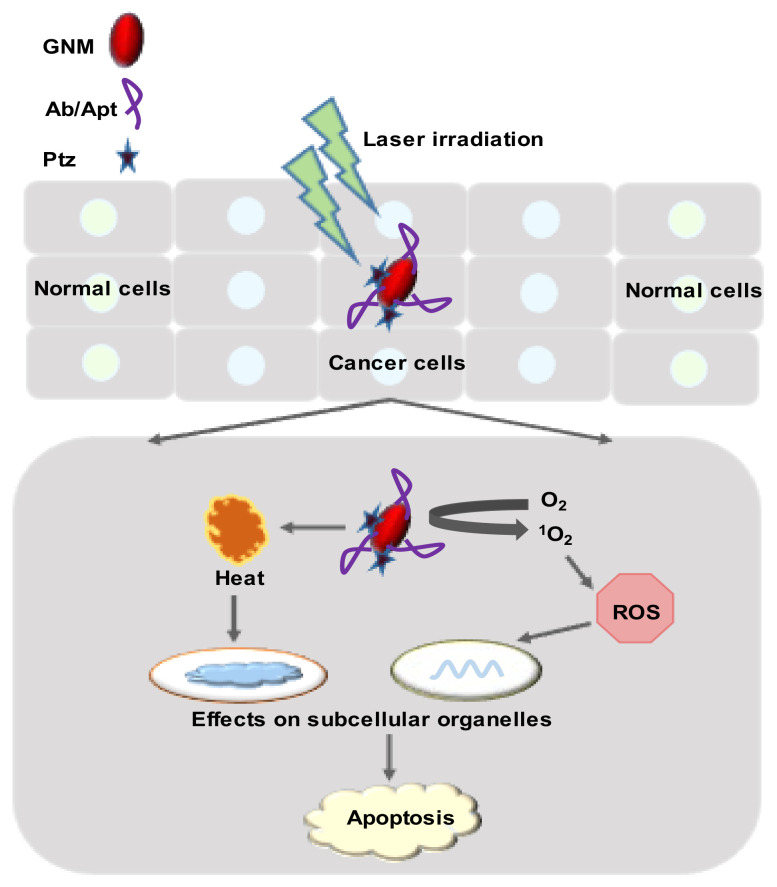
Schematic representation of GNM-mediated PTT or PDT effects. Apt- or Ab-conjugated GNM binds selectively with the cancer cells. GNM or the Ptz absorb the NIR laser light. Different cellular events involved in cell death induced by GNM-mediated PTT or PDT effects upon photoexcitation. Photothermal therapy: PTT; Photodynamic therapy: PDT; Gold nanomaterials: GNM; Near-infrared radiation: NIR; Antibody: Ab; Aptamer: Apt; Reactive oxygen species: ROS.

**Figure 8 cancers-14-03047-f008:**
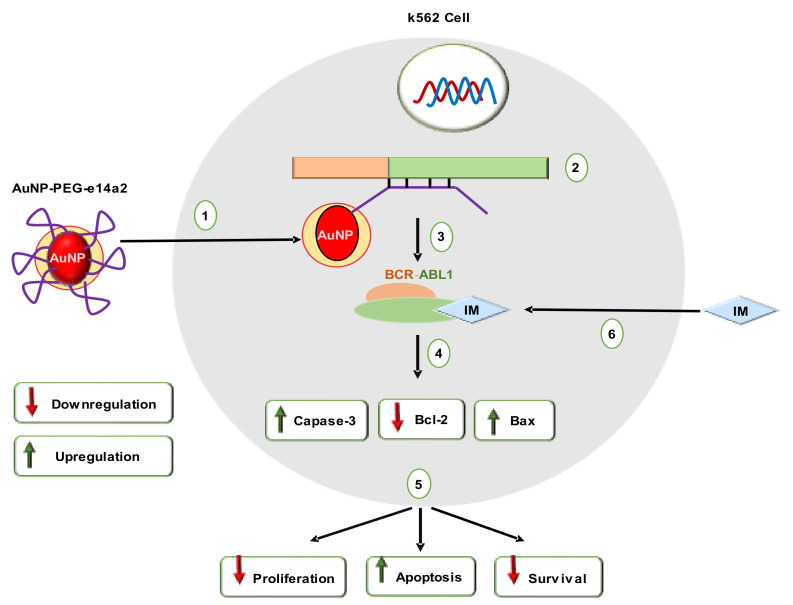
AuNPs-based BCR-ABL gene silencing. (1) AuNPs functionalized with the e14a2 antisense hairpin ssDNA oligonucleotide (AuNP-PEG-e14a2) internalized by K562 cells, a CML in vitro model. (2) The nanoconjugate recognized BCR-ABL1 mRNA and induced silenced-gene expression and triggered mRNA degradation, thereby inhibiting (3) tyrosine kinase. The nanoconjugate (4) upregulated Bax and caspase-3, while it downregulated BCL2 expression. (5) AuNP-PEG-e14a2 increased apoptosis, resulting in decreased cell proliferation and survival. (6) IM combined with AuNP-PEG--e14a2 could be used to overcome chemoresistance. The idea of this figure was reprinted from Vinhas et al. (2017) [113]. Gold nanoparticles: AuNPs; Imatinib: IM.

**Figure 9 cancers-14-03047-f009:**
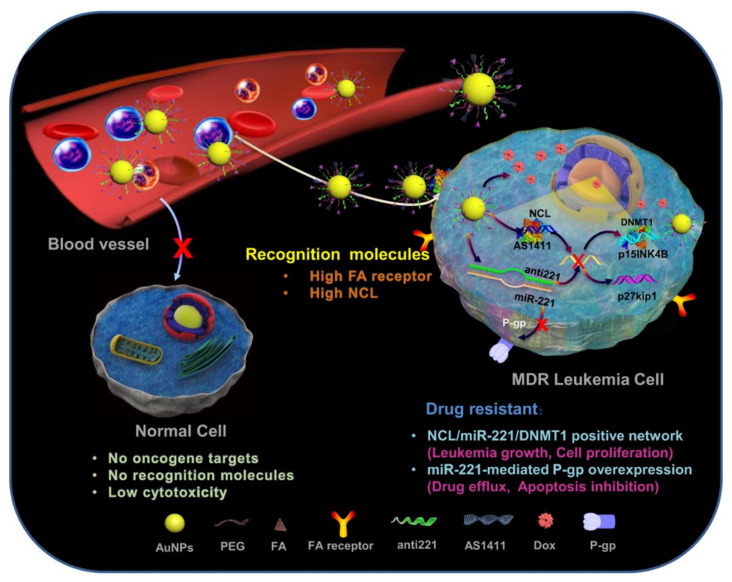
Multifunctional AuNPs target miR-221 network as well as P-gp. These multifunctional AuNPs can overcome P-gp-mediated multidrug-resistance in leukemia cells. The figure is reprinted from Deng et al. (2019) [120]. This study is under Creative Commons Attribution 4.0 International License, which permits use, sharing, adaptation, distribution, and reproduction in any medium or format, as long as you give appropriate credit to the original author(s) and the source, provide a link to the Creative Commons license (http://creativecommons.org/licenses/by/4.0/, accessed on 21 December 2021). Gold nanoparticles: AuNPs; Polyethylene glycol: PEG, Folic acid: FA; Doxorubicin: Dox; Nucleolin: NCL.

**Figure 10 cancers-14-03047-f010:**
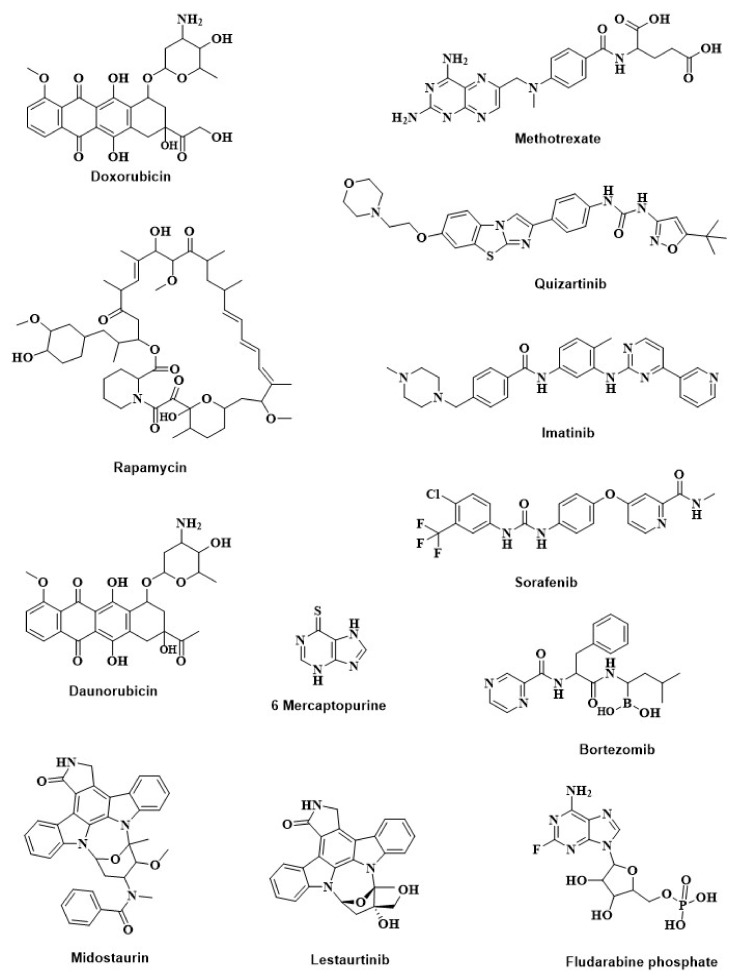
Structure of chemotherapeutic drugs used in HMs treatment. GNMs-conjugation improved the anticancer activity and selectivity of the chemotherapeutic drugs. Hematological malignancies: HMs; Gold nanomaterials: GNMs.

**Figure 11 cancers-14-03047-f011:**
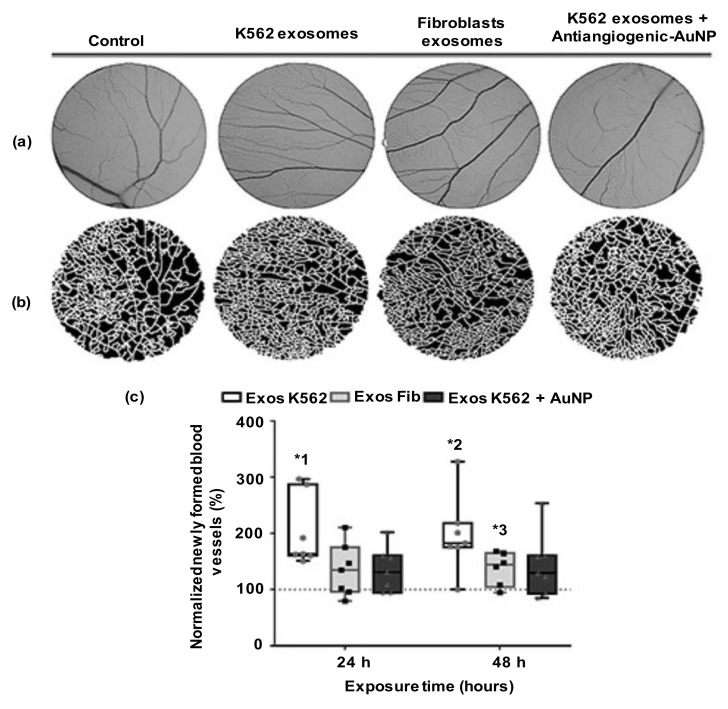
Response of antiangiogenic peptide-functionalized AuNPs in chorioallantoic-membrane model (CAM). Control areas are treated with PBS, K562 cells (50 μg/mL) exosome, fibroblasts (50 μg/mL) exosome, K562 exosomes (50 μg/mL) and antiangiogenic loaded AuNPs (16.4 nM). (**a**) CAM-region images obtained using green channel. (**b**) Represented image segment used to compute newly formed branches. (**c**) Obtained results are presented in the whisker plots. Data were obtained from six independent experiments and normalized with corresponding CAM area at 0 h incubation after PBS exposure. The 100% normalized dotted line at newly formed vessels is considered as the control sample. *1 *p*-value 0.0113, *2 *p*-value 0.0212, and *3 *p*-value 0.040 compared to control. This figure is adapted from Rodrigues et al. (2019) [86], non-commercial uses of this work do not require any further permission from Dove Medical Press Limited, under the license (http://creativecommons.org/licenses/by-nc/3.0/, accessed on 29 January 2022). Gold nanoparticles: AuNPs; Chorioallantoic membrane: CAM; Phosphate buffer saline: PBS.

**Figure 12 cancers-14-03047-f012:**
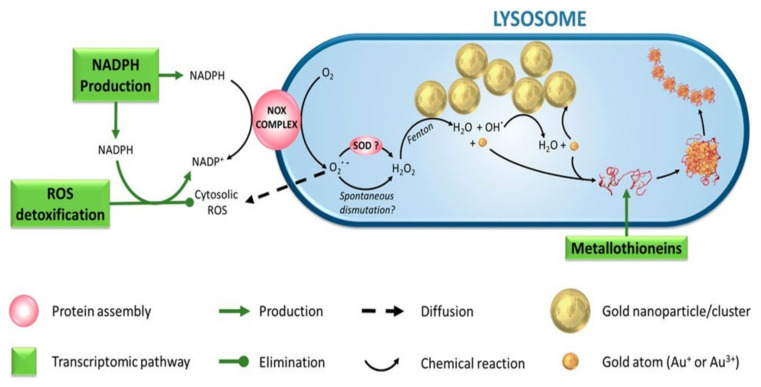
Mechanism of GNMs recrystallization and degradation process. This summary diagram of GNMs life cycle was reprinted from Balfourier et al. (2020) [203]. The mechanism of GNMs clearance was predicated based on in vitro experimental results in human-skin primary-fibroblasts cells [203]. This study is published under PNAS license (https://www.pnas.org/authors/fees-and-licenses, accessed on 22 September 2021), permission was sincerely taken from PNAS. It is noted that stoichiometric coefficients and H+ are not presented here for clarity.

**Table 1 cancers-14-03047-t001:** Using gold nanomaterials in the diagnosis of hematological malignancies.

Cancer Class	Cancer Sub-Type	GNMs	Size (nm)	Conjugated Materials	Cell Line/Test Sample	Diagnosed Cells/Biomarker	Detection Techniques	Detection Range	Reference
Leukemia	HPL	GNCs	* 26	Fe3O4, ^#^ KH1C12	HL-60	HL-60 cells	MRI, FLI	10 to 200 cells/μL	[71]
	ALL	AuNPs	NR	Fe3O4, ^#^ sgc8c	CCRF-CEM	CCRF-CEM cells	EIS	10 to 1 × 10^6^ cells/mL	[73]
	ALL	AuNPs	NR	APBA, ^#^ sgc8c	CCRF-CEM	CCRF-CEM cells	QCM, FLI	2 × 10^3^–1 × 10^5^ cells/mL	[75]
	ALL	AuNPs	15–18	Ab2	Antigen	CD10	QCM	1.0 × 10^−8^ to 1.0 × 10^−11^ M	[76]
	ALL	AuNPs	* 15	FA, FITC	CCRF-CEM	FAR	FLI	NR	[77]
	ALL	AuNPs	NR	shDNA, GPS, PCT	cDNA	BCR-ABL fusion	EIS, CHR, DPV, CV	100.0 μM to 10.0 pM	[78]
	CLL	AuNPs	60	PEG, Ab3	Antigen	CD20	SERS, DFM	NR	[79]
	CLL	AuNPs	20	PEG, Ab4	CCLP	CD19	SERS, DFM	NR	[80]
	CLL	AuNPs	60–70	shDNA, AED	cDNA	PBGD	EIS	7.0 × 10^−12^–2.0 × 10^−7^ mol/L	[72]
	AML	AuNPs	* 40–80	γ-Fe_2_O_3_, ssDNA	cDNA	WT1	SERS	NR	[24]
	CML	Pd@AuNPs	51	FA	K652	FAR	CLR	10^4^ cells/mL	[81]
	CML	AuNPs	14.6 ± 1.7	PEG, TFH	cDNA	BCR-ABL fusion	FRETS	NR	[82]
Lymphoma	NR	AuNPs	30	Ab5	K299	CD25	MPM	NR	[83]
	NR	AuNPs	40	R-Ab	Raji	CD20	SPCT	10^2^ to 10^10^ cells	[84]
Myeloma	MM	AuNPs	15	Magnetite	Myeloma Patients	PPC, HSP75	MAS	NR	[74]

Gold nanoclusters: GNCs; Gold nanoparticles: AuNPs; Human promyelocytic leukemia: HPL; an aptamer synthesized through thiolen click reaction between poly(ethylene glycol) dimethacrylate: KH1C12 aptamer; size with aptamer or coating materials’ conjugation: *; Aptamer: #; Magnetic-resonance imaging: MRI; Fluorescence imaging: FLI; Acute lymphoblastic leukemia: ALL; Folic acid: FA; Fluorescein isothiocynate: FITC; Folic-acid receptor: FAR; Aminophenylboronic acid: APBA; Electrochemical-impedance spectroscopy: EIS; Quartz-crystal microbalance: QCM; an antibody that targets CD10: Ab2; Polyethelene glycol: PEG; Not reported: NR; Surface-enhanced Raman spectroscopy: SERS; Dark-field microscopy: DMF; Chronic lymphocytic leukemia: CLL; an antibody that targets CD20: Ab3; an antibody that targets CD19: Ab4; Chronic lymphocytic leukemia isolated from patients: CLLP; Karpas 299 lymphoma cells: K229; an antibody that target CD25: Ab5; DNA containing a WT1 (XGGGCGTGTGACCGTAGCTTTAACC CTGATTGCGAATAGCG, where X = Amino C6 labeled with malachite green) sequence: ssDNA; Complementary DNA: cDNA; Wilm’s tumor gene: WT1; Palladium gold nanoparticles: Pd@AuNPs; Colorimetric: CLR; Acute myeloid leukemia: AML; Thiol-oligo-fluorophore hairpin: TFH; Förster resonance energy-transfer-based spectroscopy: FRET; –SH modified DNA: shDNA; Gold electrode: AED; Porphobilinogen deaminase gene: PBGD; Differential-pulse voltammetry: DPV; Chronoamperometry: CHR; Cyclic voltammetry: CV; Graphene sheet: GPS, poly(catechol): PCT; Multi-photon microscopy: MPM; Rituximab: R-Ab; Spectral photon-counting computed tomography (SPCT); Plasma-protease C1 inhibitor: PPC, Heat-shock protein HSP75: HSP75; Mass spectroscopy: MAS; Chronic myeloid leukemia: CML; Multiple myeloma: MM.

**Table 2 cancers-14-03047-t002:** The best AO sequence for BAG1, MDM2, Bcl-2, BIRC5 (survivin), and XIAP gene.

Gene	AO Sequence (5′–3′)	Efficacy Score
*BAG1*	UUGAAGCAGAAGAAACACU	0.99
*MDM2*	UUACAGCACCAUCAGUAGG	0.99
*BCL2*	UCAAUCUUCAGCACUCUCC	0.98
*Survivin*	UUCAAGACAAAACAAGAGC	0.97
*XIAP*	UAAGAACAACAUAACAUGC	0.97

The AOs were selected based on the efficacy score obtained from OligoWalk online software (http://rna.urmc.rochester.edu/servers/oligowalk2/help.html, accessed on 21 December 2021) against common mRNA variants of each gene. The mRNA variants sequences were obtained from the NCBI database. Results were obtained from Zaimy et al. (2016) [115]. Antisense oligonucleotide: AO.

**Table 3 cancers-14-03047-t003:** Effect of irradiating femtosecond pulses on cell fate [171].

# of Pulses	Damage Mechanism	Effect(s)
1–2	ROS	Apoptosis
3–6	ROS + cell fusion	Apoptosis, necrosis, multi-nucleic cells
7–	ROS + cell fusion + membrane rupture	Necrosis

Antibody-tagged GNHs selectively damaged the targeted cancer cells via the different mechanisms of action(s), while sparing untargeted neighboring cells. Gold nanosphere: GNHs; Reactive oxygen species: ROS.

**Table 4 cancers-14-03047-t004:** Promise of gold nanomaterials in treatment of hematological malignancies.

Treatment	HMs Type	GNMs	Size (nm)	Conjugated Materials	Tested Cell Line(s)	IC_50_	In Vivo	Upregulated Protein/ Nucleic Acid	Downregulated Protein/Nucleic Acid	Ref.
PTT	LKM	AuNPs	30	IgG	K562	NR	NR	NR	NR	[3]
	LKM	AuNPs	30	MAB1, MAB2	^+^ LKM	NR	NR	NR	NR	[93]
	LKM	GNR	NR	CD33, PEG	HL-60, K-562	NR	NR	NR	NR	[94]
	LYP	AuNPs	NR	anti-CD138-Ab	A20	NR	NR	NR	IgG2a	[96]
PDT	LKM	GNR	NR	Ce6, sgc8c aptamer	CCRF-CEM, Ramos	NR	NR	NR	NR	[103]
	LKM	AuNPs	45	Ce6	Jurkat, Jurkat/A4	NR	NR	NR	NR	[104]
	LKM	AuNPs	16	5-ALA	K562	NR	NR	NR	NR	[106]
RDT	LKM	^#^ AuNPs	* 22 ± 2	PEG	HL-60 II, Jurkat D1.1	NR	NR	NR	NR	[108]
GNT	LKM	AuNPs	14	PEG, e14a2	K562	IM (22 mM), IMA (17 mM)	NR	Bax, Caspase-3	BCR-ABL1, Bcl2	[113]
	LKM	AuNPs	* <50	AOs, anti-Apt, Dox	AML-M2	>150 μg/mL	NR	NR	BCL-2, BAG1, MDM2, BIRC5, XIAP	[115]
	LKM	AuNPs	13	NLS, AS1411, anti-221	Jurkat, Kasumi-1, K562, HL60, NB4, Thp1, Molt4, 293 T, U937, C1498	NR	C57BL/6 mice	p15INK4B, p27kip1	miR-221, DNMT1	[119]
	LKM	AuNPs	40	FA, AS1411, anti-221, Dox	Drug resistant K562, AML RP1, AML RP1, AML RP3	0.56 μM (DR K562), 0.31 μM (AML RP1), 0.53 μM (AML RP2), 0.08 μM (AML RP3)	NR	p15INK4B, p27kip1	miR-221, DNMT1, P-gp	[120]
DCT	LKM	AuNPs	5	Anti-Tim-3-ScAb, RAP	THP-1	NR	NR	NR	p-eIF4E-BP	[26]
	LKM	AuNPs	20	FLP, FA	KG1	<2 mM	NR	NR	NR	[124]
	LKM	AuNPs	5	MPA, As2O3	K562, KA	~2.2 × 10^−2^ mg/L (K562), ~1.4 × 10^−2^ mg/L (KA)	NR	NR	NR	[125]
	MM	AuNPs	∼5	VEL, FA	RPMI, U226	NR	NR	NR	NR	[126]
	LKM	m-AuNPs	30–40	Dox	HL-60 and K562	NR	NR	NR	NR	[128]
	LKM	AuNPs	15.2 ± 0.7	Dau, sgc8c aptamer	Molt-4, U266	~5 μM (Molt-4), >5 μM (U266)	NR	NR	NR	[130]
	LKM	AuNPs	~12	PLU, GEL, SOR, LES, QUI	THP1, OCI-AML3	NR	NR	NR	FLT3	[134]
	LKM	AuNPs	17 ± 2	PLU, MDS	THP1, OCI-AML3	NR	NR	NR	NR	[136]
	LKM	AuNPs	~17	IM, Topo, CBT	K562	NR	NR	NR	NR	[138]
	LKM	AuNPs	4–5	6-MTP	K-562	NR	NR	NR	NR	[141]
	LKM	AuNPs	~2.5	MTX	TPH-1	NR	NGS mice	NR	NR	[144]
ABT	MM	AuNPs	26 ± 7	AbMM	SP2OR	NR	Mice	p21, p27	NR	[148]
	LKM	AuNPs	NR	R-Ab	HS 505.T, CLL-AAT	NR	NR	NR	MS4A1, CD20	[150]
	LYP	AuNPs	20	R-Ab	Raji	NR	NR	NR	BCL-2	[153]
	LYP	AuNPs	30	PEG, R-Ab	Z138	NR	NR	NR	NR	[154]
PPT	LKM	AuNPs	3 ± 2	AP, OEG	K562	NR	NR	NR	VEGFR1	[86]
BCT	LKM	AuNPs	15–30	*B. serrata* LE	HL-60/vcr, 32D-FLT3-ITD, Murine C1498	329 μg/mL (HL-60/vcr) 320 μg/mL (32D-FLT3-ITD), 219 μg/mL (Murine C1498)	DMBA mice	IFNα, IL4, IL5, IL10, IL13, IFNα, S1PR1 and S1PR5 mRNA	IFNY, TNFα, IL1, IL6, IL12, and IL18	[157]
	LKM	AuNPs	10–40	*L. culinaris* SE	HL-60/vcr, 32D-FLT3-ITD, Murine C1498	246 μg/mL (HL-60/vcr) 367 μg/mL (32D-FLT3-ITD), 212 μg/mL (Murine C1498)	DMBA mice	IFNα, IL4, IL5, IL10, IL13, IFNα, S1PR1, S1PR5 mRNA	IFNY, TNFα, IL1, IL6, IL12, and IL18	[158]
	LKM	AuNPs	20–30	*C. sinesis* LE	HL-60/vcr, 32D-FLT3-ITD, Murine C1498	224 μg/mL (HL-60/vcr) 258 μg/mL (32D-FLT3-ITD), 158 μg/mL (Murine C1498)	DMBA mice	IFNα, IL4, IL5, IL10, IL13, IFNα	IFNY, TNFα, IL1, IL6, IL12, and IL18	[159]
	LKM	AuNPs	10–30	*T. vulgaris* LE	HL-60/vcr, 32D-FLT3-ITD, Murine C1498	218 μg/mL (HL-60/vcr) 336 μg/mL (32D-FLT3-ITD), 186 μg/mL (Murine C1498)	DMBA mice	IFNα, IL4, IL5, IL10, IL13, IFNα, S1PR1, S1PR5 mRNA	IFNY, TNFα, IL1, IL6, IL12, IL18	[161]
	LKM	AuNPs	15–45	*H. sabdariffa* FE	HL-60/vcr, 32D-FLT3-ITD, Murine C1498	189 μg/mL (HL-60/vcr) 309μg/mL (32D-FLT3-ITD), 185 μg/mL (Murine C1498)	DMBA mice	IFNα, IL4, IL5, IL10, IL13, IFNα, S1PR1, S1PR5 mRNA	IFNY, TNFα, IL1, IL6, IL12, IL18	[160]
RST	LYP	GNHs	20	R-Ab	BJAB, K562	NR	NR	NR	NR	[171]
	LKM	GNR	* 122 ± 1	USPN	K562	NR	NR	Beclin-1	BCR-ABL, p-PI3K, p-AKT	[172]
APT	LKM	AuNPs	* 10	PEG	K562	<10 mM	NR	NR	NR	[173]
	LKM	AuNPs	4	VF-Ab	^+^ CLL B	NR	NR	Cleaved PARP	Mcl-1, BcL-2 and caspase3	[174]
	LKM	GNSs	2–5	VER, PNM	KA	NR	Nude mice	GAPDH; Cleaved Caspase-3, 8, and 9	NR	[176]
	LYP	AuNPs	15	tmCpG	A20, Ramos, JeKo-1, Mino, RC, REC-1, DLBCL (SUDHL4)	NR	BALB/c mice	IL-6, TNFα, CD19, CD20, CD47	NR	[177]
	LKM	AuNPs	<10	*S. muticum*	K562, Jurkat, HL-60, CEM-ss cells	4.22 ± 1.12 (K562), 5.71 ± 1.4 (HL-60), 6.55 ± 0.9 (Jurkat), 7.29 ± 1.7 μg/mL (CEM-ss)	NR	Caspase-3, caspase-9	NR	[178]
	LYP	AuNPs	16	*O. sanctum* LE	DL	>50 μg/mL	NR	NR	NR	[179]
	LKM	AuNPs	3	Asn	CCRF-CEM	NR	NR	NR	MMP-2	[180]
	LKM	GNR	5.55 ± 1.56	SI, FA	EL4s	<75 μM	NMRI mice	NR	NR	[182]
Others	LKM	AuNPs	5	NR	OPM-1, RPMI-8266, U-266	< 20 μg (OPM-1, RPMI-8266), >20 μg (U-266)	NR	p21, p27	NR	[149]

Hematological malignancies: HMs; Gold nanomaterials: GNMs; References: Ref.; Photothermal therapy: PTT; Leukemia: LKM; Gold nanoparticles: AuNPs; Secondary IgG antibody: IgG; Monoclonal antibodies: MAB; Primary monoclonal antibody for CD-10, CD-19, and CD-20: MAB-1; Secondary monoclonal antibody for CD-10, CD-19, and CD-20: MAB-2; Gold nanorod: GNR; Polyethylene glycol: PEG; Lymphoma: LYP; Antibody: Ab; Photodynamic therapy: PDT; Chlorin e6: Ce6; Subline of Jurkat cells with multidrug resistance phenotype: Jurkat/A4; 5-aminolevulinic acid: 5-ALA; Radiotherapy: RDT; Gene therapy: GNT; Abelson murine leukemia (ABL1) gene located on chromosome nine with the breakpoint cluster region (BCR) gene: BCR-ABL1; Imatinib: IM; Palindromic sequence 5′-TTTCGGCGCTGAAGGGCTTTTGAACTCCGAAA-3′ targeting the fusion e14a2 BCR-ABL1 transcript: e14a2; IM+ AuNPs-PEG-e14a2: IMA; Antisense oligonucleotides: AOs; anti-CD33(+)/CD34(+) aptamer: anti-Apt; Doxorubicin: Dox; Nuclear localization signal peptide: NLS; A 26-nucleotide DNA aptamer: AS1411; an antisense oligonucleotide: anti-221; DNA Methyltransferase 1: DNMT1; Folic acid: FA; Drug resistant K562: DR K562; Acute myeloid leukemia (AML) patient-1 derived AML relapse cells: AML RP1; AML patient-2 derived AML relapse cells: AML RP2; AML patient-3 derived AML relapse cells: AML RP3; Methotrexate: MTX; Non-obese diabetic severe combined immunodeficiency gamma mice: NSG mice; Delivery of chemotherapeutic drugs: DCT; Daunorubicin: Dau; Anti-Tim-3 single-chain antibody: Anti-Tim-3-ScAb; Phosphorylated eukaryotic initiation factor 4E binding protein: p-eIF4E-BP; Rapamycin: RAP; Fludarabine Phosphate: FLP; 3-mercaptopropionic acid: MPA; Multiple myeloma: MM; Adriamycin-resistant K562/A02 cell line: KA; Velcade: VEL; Monodisperse AuNPs: m-AuNPs; Pluronic: PLU; Gelatin: GEL; Sorafenib: SOR; Lestaurtinib: LES; Quizartinib: QUI; Midostaurin: MDS; Topotecan: Topo; 4-carboxylic benzoic acid linker: CBT; 6-mercaptopurine-9-ß-D-ribofuranoside: 6-MTP; p-glycoprotein: P-gp; Arsenic trioxide: As2O3; Antibody-based targeted therapy: ABT; Polyclonal Antimyeloma antibody: AbMM; Rituximab: R-Ab; Membrane-spanning 4-domains, subfamily A, member 1: MS4A1; Peptide-based antiangiogenic therapy: PPT; Vascular endothelial growth factor receptor 1: VEGFR1; Antiangiogenic peptide: AP; Oligo ethylene glycol: OEG; Bioactive compound-based therapy: BCT; Leaf extract: LE; 7,12-dimethylbenz[a]anthracene induced AML mice: DMBA mice; Seed extract: SE; Flower extract: FE; Camellia sinesis: C. sinesis; Boswellia serrata: B. serrata; Sphingosine-1-phosphate receptor-1: S1PR1; Sphingosine-1-phosphate recep-tor-5: S1PR5; Lens culinaris: L. culinaris; Thymus vulgaris: T. vulgaris; Hibiscus sabdariffa: H. sabdariffa; Reactive oxygen species-based therapy: RST; Gold nanosphere: GNHs; Ultra-small platinum nanoparticles: USPN; Apoptosis-induction therapy: APT; anti-VEGF antibody: VF-Ab; Gold nanoshells: GNSs; Verbascoside: VER; poly-(N-isopropyl acrylamide): PNM; Cytosine-phosphate-guanine (CpG) sequences modified with tri-ethylene glycol: tmCpG; Asparagine: Asn; Silica: Si; Ocimum sanctum: O. sanctum; Matrix metallo- proteinase-2: MMP-2; High-grade B cell lymphoma cells: RC; Size with coating materials or drug conjugation: *; Patient derived cells: ^+^; Radionazation treatment: ^#^.

## Data Availability

Data reported in Figure 2 are available via the cBioportal database (https://www.cbioportal.org/, 29 January 2022).

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
