# Peer review of "Theranostic Potentials of Gold Nanomaterials in Hematological Malignancies"

_cancers, 2022, doi:10.3390/cancers14133047_

Round 1

Reviewer 1 Report

The review is well written and touches on a current topic. The material presented will be of interest to readers of Cancers journal. However, there are several recommendations to improve the overview:
1) In the introduction, it should be noted whether there were reviews of a similar content and what is the relevance of this review.
2) In the introduction, it is necessary to indicate the principle by which the cited literature was selected, as well as the time period that the review covers.
3) The conclusions should be somewhat expanded.

Author Response

Reviewer 1:

The review is well written and touches on a current topic. The material presented will be of interest to readers of Cancers journal. However, there are several recommendations to improve the overview:

Thank you very much for your insightful comments. Here are the point-by-point responses.

Point 1: In the introduction, it should be noted whether there were reviews of a similar content and what is the relevance of this review.

Response 1: One recent review published which its relevant to our manuscript. We have added it in the introduction part. Please see the line number 124-126. Thank you.

Point 2: In the introduction, it is necessary to indicate the principle by which the cited literature was selected, as well as the time period that the review covers.

Response 2: Article searching strategies have been added in the introduction according to your suggestions. Please see the line number 126 -133. Thank you.

Point 3: The conclusions should be somewhat expanded.

Response 3: Conclusions part has been expanded. Please see the line number  1197-1209. Thank you.

Reviewer 2 Report

The authors provided a well-documented overview of the potential application of gold nanomaterials in hematological malignancies. The field of research focused on nanomedicine applied on HMs is in continuous evolution and even if the article is well written, the introduction section miss a key actor in the B cell HMs: the IgG! IgGs together  whit the complex mechanisms of IgG rearrangment represent an emerging target for patient stratification (PMID: 32483301), tumor progression monitoring and exosome-based application for MRD identification (PMID: 32483301). In definitive, the review could represent a really interesting point of view in the perspective of future nanotechnological platform definitions. 

I hope that my comments and suggestions could be useful and I look forward to reading the revised version of the paper.

Good luck.

Author Response

Reviewer 2:

Point 1: The authors provided a well-documented overview of the potential application of gold nanomaterials in hematological malignancies. The field of research focused on nanomedicine applied on HMs is in continuous evolution and even if the article is well written, the introduction section miss a key actor in the B cell HMs: the IgG! IgGs together whit the complex mechanisms of IgG rearrangment represent an emerging target for patient stratification (PMID: 32483301), tumor progression monitoring and exosome-based application for MRD identification (PMID: 32483301). In definitive, the review could represent a really interesting point of view in the perspective of future nanotechnological platform definitions. 

I hope that my comments and suggestions could be useful and I look forward to reading the revised version of the paper.

Good luck.

Response 1: Thank you very much for your appreciation and insightful comments. Suggested reference is interesting, and we have added it in the introduction part. Please see the line number 81-83. Thank you.

Reviewer 3 Report

Major Inconsistencies:

  • In the references lists, the number that refereed to the publications or websites is missing making impossible to verify the information. I cannot in this case review correctly the review. Clearly the authors have not performed a serious proofreading of the review.
  • LN 20 “Hematological malignancies cover 50% of all malignancies”. I would rather say Hematologic malignancies than hematological malignancies. Second, 50% is clearly not true. HM incidence is 6.5% of all cancers around the world, including approximately 9.0% in the United States and Europe.
  • In the abstract, the use of GNM as theranostic agent is weirdly introduce. LN 45. “GNMs may cause irreversible damage […] through reactive oxygen species”. GNM does not produce ant reactive oxygen species by its own. X ray coupled with GNM induce ROS production. Also GNM is not by itself targeting HM. Vectorized GNM can do it. Some important information is missing.

Author Response

Reviewer 3:

Major Inconsistencies:

Point 1: In the references lists, the number that refereed to the publications or websites is missing making impossible to verify the information. I cannot in this case review correctly the review. Clearly the authors have not performed a serious proofreading of the review.

Response 1: We have re-checked the reference order, cited publications, and websites corresponding to the text. We did not find any inconsistencies in reference section corresponding to the text. We believe there were some misunderstandings. During writing this review article, we carefully read and the cited each of the papers. We welcome the reviewer to have a look at the manuscript once again. Thanks for your comment.

Point 2: LN 20 “Hematological malignancies cover 50% of all malignancies”. I would rather say Hematologic malignancies than hematological malignancies. Second, 50% is clearly not true. HM incidence is 6.5% of all cancers around the world, including approximately 9.0% in the United States and Europe.

Response 2: In our manuscript, it’s written that “HMs comprise about 50% of all malignancies in children and 5-8% of adult malignancies [1]”. It indicates that about 50% of all malignancies in children and 5-8% in adults. Please kindly check the introduction part of the cited reference for further clarification. Thank you.

Reference: Lapotko, D.; Lukianova, E.; Shnip, A.; Zheltov, G.; Potapnev, M.; Savitsky, V.; Klimovich, O.; Oraevsky, A. Laser activated nanothermolysis of leukemia cells monitored by photothermal microscopy. In Proceedings of the Photons Plus Ultrasound: Imaging and Sensing 2005: The Sixth Conference on Biomedical Thermoacoustics, Optoacoustics, and Acousto-optics, 2005; pp. 82-89.

Point 3: In the abstract, the use of GNM as theranostic agent is weirdly introduce. LN 45. “GNMs may cause irreversible damage […] through reactive oxygen species”. GNM does not produce ant reactive oxygen species by its own. X ray coupled with GNM induce ROS production. Also GNM is not by itself targeting HM. Vectorized GNM can do it. Some important information is missing.

Response 3: Thank you very much for your comment. Yes. we agree with the reviewer, and we have edited the abstract and related information in the manuscript according to your and other reviewer’s comments. Please see the line number 42-56, 938-941. Thank you.

Reviewer 4 Report

This Review manuscript summarizes the roles of gold nanomaterials in hematological malignancies diagnosis and treatment. Different detection methods are displayed in detail. Gold nanomaterial-based treatments in hematological malignancies including photothermal therapy, photodynamic therapy, radiation therapy, gene therapy, etc. are exhibited point for point. In addition, abundant references are provided in this Review manuscript.

I think it’s a timely, interesting, and important Review. I suggest some minor revision for this manuscript.

1. In the Simple Summary, add one sentence to highlight the uniqueness of gold nanomaterials from quantum chemistry perspective or other viewpoint.

2. In the Abstract, pay more attention to the structures and functions of gold nanomaterials. Simplify the analysis of traditional and commercial hematological malignancies diagnostic and therapeutic strategies. Add more analysis about the applications of gold nanomaterials in hematological malignancies diagnosis and treatment.

3. In the Introduction, add one Figure about the structures of different gold nanomaterials and corresponding applications in hematological malignancies diagnosis and treatment. Some references are recommended, such as Frontiers in Bioscience-Landmark 2022, 27(2), 40; Biosensors and Bioelectronics 2020, 150, 111869.

4. Add one Figure to explain the molecular mechanism of the formation of Hematological malignancies.

5. Merge Figure 1 and Figure 2 and simplify the captions.

6. There is one obvious mistake. There are two sections of 3.2.2. Detection of receptor overexpression.

7. Using 3.2.3 Nucleic acids biomarker detection as a replacement of 3.2.3. DNA biomarker detection.

8. I think Table 1 is a comprehensive summary of gold nanomaterials in the diagnosis of hematological malignancies. Put the Table 1 in the front of this manuscript.

9. Explain the mechanism of reactive oxygen species-mediated cytotoxicity for hematological malignancies therapy using gold nanomaterials as the catalysts.

10. Provide some possible solutions to tackle the Challenges of using GNMs in HMs.

Author Response

Reviewer 4:

This Review manuscript summarizes the roles of gold nanomaterials in hematological malignancies diagnosis and treatment. Different detection methods are displayed in detail. Gold nanomaterial-based treatments in hematological malignancies including photothermal therapy, photodynamic therapy, radiation therapy, gene therapy, etc. are exhibited point for point. In addition, abundant references are provided in this Review manuscript.

I think it’s a timely, interesting, and important Review. I suggest some minor revision for this manuscript.

Thank you very much for your appreciation, and insightful comments. Here are our point-by-point responses.

Point 1: In the Simple Summary, add one sentence to highlight the uniqueness of gold nanomaterials from quantum chemistry perspective or other viewpoint.

Response 1: We have added uniqueness of gold nanomaterials from quantum chemistry perspective in the simple summery. Please see the line number 26-27. Thank you.

Point 2: In the Abstract, pay more attention to the structures and functions of gold nanomaterials. Simplify the analysis of traditional and commercial hematological malignancies diagnostic and therapeutic strategies. Add more analysis about the applications of gold nanomaterials in hematological malignancies diagnosis and treatment.

Response 2: We have revised the abstract according to your suggestions. Please see the line number 32-61. Thank you.

Point 3: In the Introduction, add one Figure about the structures of different gold nanomaterials and corresponding applications in hematological malignancies diagnosis and treatment. Some references are recommended, such as Frontiers in Bioscience-Landmark 2022, 27(2), 40; Biosensors and Bioelectronics 2020, 150, 111869.

Response 3: We have added a figure (Figure 1.) showing structures of different gold nanomaterials. We do apologise as we did not add their corresponding applications in hematologic malignancies. All the corresponding applications are mentioned in the summery Table-1,2. Hope your consideration. Additionally, referred papers have been added in the revised manuscript. Please see the line number 124-126, 1200-1202 . Thank you.

Point 4: Add one Figure to explain the molecular mechanism of the formation of Hematological malignancies.

Response 4: We have drawn a simple figure (Figure 3.) showing the molecular mechanism of the formation of hematological malignancies, we hope it’s alright now. Thank you.

Point 5: Merge Figure 1 and Figure 2 and simplify the captions.

Response 5:  We merged previous Figure 1 and Figure 2 and simplified the caption. In the revised manuscript it’s Figure 2, we hope it’s alright now. Thank you.

Point 6: There is one obvious mistake. There are two sections of 3.2.2. Detection of receptor overexpression.

Response 6:  We apologize for this typo error. We have corrected it in the revised manuscript. Please see the subsection 3.2.2. and 3.2.3.  Thank you

Point 7: Using 3.2.3 Nucleic acids biomarker detection as a replacement of 3.2.3. DNA biomarker detection.

Response 7:  We have revised the subsection according to your suggestion. Please see the subsection 3.2.4. Thank you.

Point 8: I think Table 1 is a comprehensive summary of gold nanomaterials in the diagnosis of hematological malignancies. Put the Table 1 in the front of this manuscript.

Response 8:  Table 1 has been ordered according to your suggestion in the revised manuscript. Thank you.

Point 9: Explain the mechanism of reactive oxygen species-mediated cytotoxicity for hematological malignancies therapy using gold nanomaterials as the catalysts.

Response 9: Mechanism of reactive oxygen species-mediated cytotoxicity for hematological malignancies therapy using gold nanomaterials as the catalysts has been added in the revised manuscript under section 4.6. Please see the line number 931-941. Thank you.

Point 10: Provide some possible solutions to tackle the Challenges of using GNMs in HMs.

Response 10:  Some of the solutions are proposed in the conclusions segment, please see the line number 1197-1209. Thank you.

Round 2

Reviewer 1 Report

In this form, the article can be published

Reviewer 3 Report

The manuscript doesn't reach the expected level... The answers of the authors do not show any improvement on the manuscript.